# The Mi-2 nucleosome remodeler and the Rpd3 histone deacetylase are involved in piRNA-guided heterochromatin formation

Bruno Mugat[1], Simon Nicot[1], Carolina Varela-Chavez[1], Christophe Jourdan[1], Kaoru Sato[2], Eugenia Basyuk[1], François Juge [3], Mikiko C. Siomi[2], Alain Pélisson[1] & Séverine Chambeyron [1✉]

In eukaryotes, trimethylation of lysine 9 on histone H3 (H3K9) is associated with transcriptional silencing of transposable elements (TEs). In drosophila ovaries, this heterochromatic repressive mark is thought to be deposited by SetDB1 on TE genomic loci after the initial recognition of nascent transcripts by PIWI-interacting RNAs (piRNAs) loaded on the Piwi protein. Here, we show that the nucleosome remodeler Mi-2, in complex with its partner MEP-1, forms a subunit that is transiently associated, in a MEP-1 C-terminus-dependent manner, with known Piwi interactors, including a recently reported SUMO ligase, Su(var)2-10. Together with the histone deacetylase Rpd3, this module is involved in the piRNA-dependent TE silencing, correlated with H3K9 deacetylation and trimethylation. Therefore, drosophila piRNA-mediated transcriptional silencing involves three epigenetic effectors, a remodeler, Mi-2, an eraser, Rpd3 and a writer, SetDB1, in addition to the Su(var)2-10 SUMO ligase.

[1] Institute of Human Genetics, UMR9002, CNRS and Univ. Montpellier, Montpellier, France. [2] Department of Biological Sciences, Graduate School of Science, The University of Tokyo, Tokyo, Japan. [3] Institut de Génétique Moléculaire de Montpellier, University of Montpellier, CNRS, Montpellier, France. ✉email: severine.chambeyron@igh.cnrs.fr

Eukaryotic genomes contain large amounts of repetitive DNA elements, including transposable elements (TEs). These sequences are major targets for the assembly of heterochromatin[1–4]. Some posttranslational modifications (PTMs) of histones, such as hypoacetylation of histone tails and trimethylation of lysine 9 on histone H3 (H3K9me3), are heterochromatin hallmarks and key players in heterochromatin assembly[5–8]. These histone PTMs allow the binding to nucleosomes of specific proteins, known as histone PTM "readers." Such readers can belong to complexes that contain or recruit chromatin-modifying enzymes that, in turn, add histone PTMs (i.e., writers) or remove them (i.e., erasers). Writers and erasers can also be recruited via interaction with sequence-specific DNA binding proteins, as described in Saccharomyces cerevisiae[9,10]. Alternatively, base-pairing between a nascent transcript and a small non-coding RNA, which is associated with an Argonaute protein in a RNA-induced transcriptional silencing (RITS) complex, is another way to specifically recruit writer/eraser proteins to sequences to be heterochromatinized.

Small non-coding RNA-mediated heterochromatin formation is best understood in Schizosaccharomyces pombe, in which the Argonaute 1 protein, guided by a small interfering RNA (siRNA), targets the complementary sequence of centromeric repeat nascent RNAs to recruit the writer Clr4 that deposits the H3K9me2 repressive mark[11–14]. H3K9me2 is then read by the chromodomain of Chp2, which recruits the Snf2/HDAC repressive complex (SHREC), a functional homolog of the nucleosome remodeling and histone deacetylase (NuRD) metazoan complex, which acts at the same time as a chromatin remodeler and an eraser, removing histone acetylation, to eventually reinforce centromeric repeat repression. These nucleosome remodeling and deacetylase sub-entities can function both independently of the whole SHREC and coordinately as the holo-complex[15,16]. In Drosophila melanogaster, the Mi-2 nucleosome remodeling ATPase resides within two different complexes, the canonical dNuRD and the Drosophila MEP-1 containing complex (dMec). As opposed to dNuRD, dMec functions independently of the HDAC subunit to act as a small ubiquitin-related modifier (SUMO)-dependent co-repressor[17]. None of these chromatin remodeler complexes has ever been described in the Drosophila TE heterochromatinization pathway.

In Drosophila ovaries, nascent TE transcripts are transiently targeted by PIWI-interacting small RNAs (piRNAs) that are associated with the Piwi Argonaute protein in a piRNA-containing RITS (piRITS) complex[18–20]. How recognition of nascent TE RNA by the piRITS leads to H3K9me3 deposition and TE heterochromatinization is still not well understood. Genetic and biochemical studies suggest that the methyltransferase SetDB1/Eggless is the writer that deposits the H3K9me3 repressive mark[21–24]. SetDB1 was recently suggested to physically and genetically interact with a Piwi interactor, the SUMO ligase Su(var)2-10, providing the first insight into its Piwi-mediated chromatin recruitment[24]. Gametocyte-specific factor 1 (Gtsf1, also known as Asterix), Panoramix (Panx, also known as Silencio), and Nxf2 have also been described to interact with Piwi[21,23,25–30]. In the experiments performed to determine the piRITS interactome, using either Piwi or its interactors Panx and Nxf2 as baits, no proteins belonging to a chromatin repressor complex that could be functionally homologous to SHREC of S. pombe have been identified.

Here we identify three piRITS corepressors in Drosophila: the Rpd3 histone deacetylase, the Mi-2 nucleosome remodeler, and its MEP-1 partner. We demonstrate the direct involvement of each of them in the transcriptional silencing of piRNA targets and in the H3K9 trimethylation and deacetylation of TE chromatin in ovarian somatic cells (OSCs). Unexpectedly, Rpd3 and Mi-2 are not conveyed to piRITS as members of the SHREC-like canonical NuRD complex. Instead, most of the somatic ovarian Mi-2 pool is associated with its MEP-1 partner in a dMec module that seems to transiently interact with a Piwi/Gtsf1 subunit. A physical link between these two modules might be provided by Su(var)2-10 that we found to interact with MEP-1 in both S2 and OSCs, in addition to its previously reported Piwi and SetDB1 partners[24]. Of note, when artificially tethered to DNA, MEP-1 is sufficient to repress the target, concomitantly with H3K9me3 deposition and H3K9 deacetylation. Together, these findings suggest that conserved mechanisms of RNA-mediated transcriptional silencing use nucleosome remodelers, chromatin erasers, and writers.

## Results

**Three known co-repressors physically interact with Piwi.** To define the interactome of Piwi, we used its Gtsf1 partner as a bait in nuclear lysates of cultured OSCs[31]. We first performed pull-down experiments using recombinant GST-Gtsf1 (ref. [25]) (Supplementary Fig. 1a), followed by mass spectrometry analysis (see Supplementary Information). This approach was validated by a $2.6 \times 10^8$-fold Piwi enrichment in GST-Gtsf1 pulldown compared with the GST control, as expected from previously described Gtsf1–Piwi interactions[25,26] (Supplementary Fig. 1b and Supplementary Data 1). Among the identified Gtsf1 partners, we decided to focus on the Rpd3 histone deacetylase (HDAC1 homolog), the Mi-2 nucleosome remodeling ATPase, and the MEP-1 Krüppel-type zinc-finger protein ($1.3 \times 10^6$-, $7.3 \times 10^7$-, and $3.8 \times 10^6$-fold enrichments as compared with GST alone, respectively), because they all belong to nucleosome remodeling and/or deacetylase silencing complexes as reported previously[32–36] (Supplementary Fig. 1b and Supplementary Data 1). Moreover, one of these three putative Gtsf1 interactors, MEP-1, had also been reproducibly identified in two previous genetic screens for factors involved in TE repression in ovarian somatic and germinal tissues[37,38].

To validate the physical links between Gtsf1 and its three putative interactors (Rpd3, Mi-2, and MEP-1), we overexpressed a green fluorescent protein (GFP)-tagged Gtsf1 protein (Gtsf1-GFP) in OSCs. We found that the Gtsf1-GFP immunoprecipitation pulled down not only Piwi but also its three putative interactors Mi-2, MEP-1 and Rpd3, as well as the Rpd3-associated protein p55 (RdAp48 homolog) (Fig. 1a). To further analyze these interactions, we used a semi-quantitative luminescence-based co-immunoprecipitation (co-IP) method (LUMIER)[39]. In this assay, Gtsf1 is fused to the FLAG-tagged Firefly luciferase (FFL) and used as a bait to quantify its interaction with various preys fused to Renilla luciferase (RL) (Fig. 1b). Following transient co-expression in S2 cells, the FFL-bait was immunoprecipitated with an anti-FLAG antibody, and the RL and FL luciferase activities were measured in the input and the IP (Fig. 1c). Luciferase activities ratio after IP gives a sensitive readout of the interaction between the bait and the prey. We used mCherry-RL as a negative prey control for co-IP. By this approach, we confirmed the strong interaction between Gtsf1 and Piwi as previously reported[25,26]. We also detected significant interactions between Gtsf1 and MEP-1, Mi-2, and Rpd3.

To prevent artificial interactions due to overexpression, we next tagged the C-terminus of the endogenous Gtsf1 in OSCs with FLAG-HA(3×) (Gtsf1-FH) using the CRISPR-Cas9 approach (Supplementary Fig. 2a, b). For unknown technical reasons, we were unable to specifically immunoprecipitate the Gtsf1 partners with anti-HA antibody. Indeed, Piwi was pulled down even in the negative control immunoprecipitation (IP) using nuclear lysates of OSC cells that do not express Gtsf1-FH (Supplementary Fig. 2c). To circumvent this problem, we used an

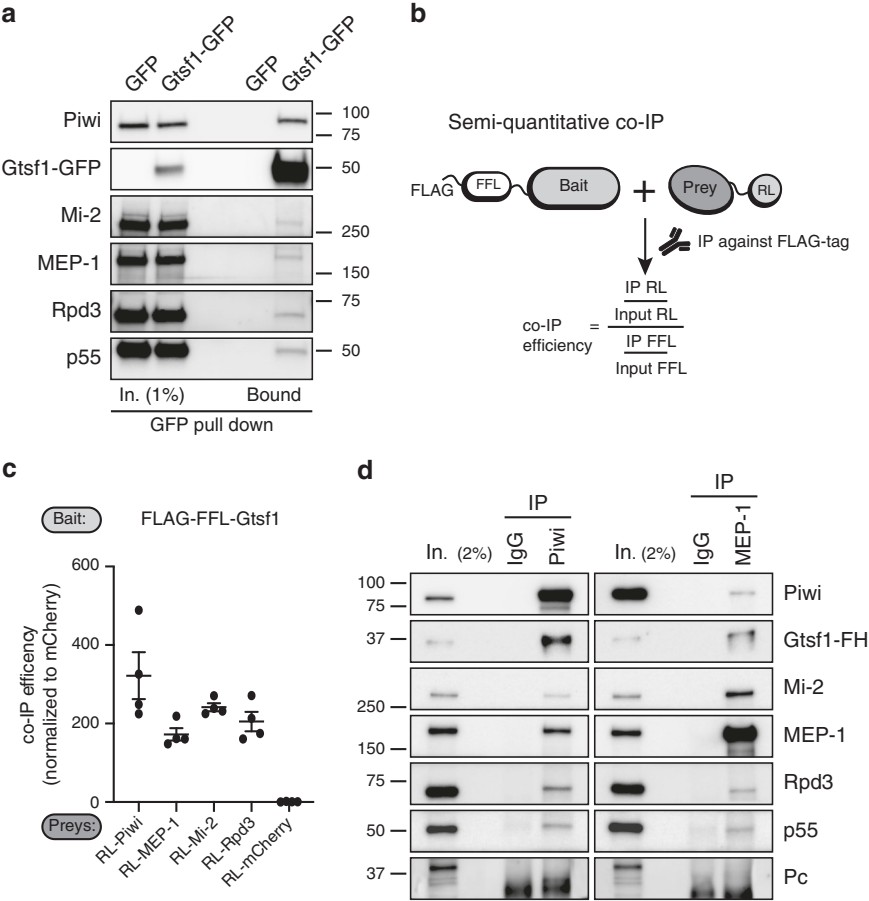

**Fig. 1 Piwi and Gtsf1 interact with MEP-1, Mi-2, Rpd3 and p55 in OSCs. a** Western blot analysis of co-immunoprecipitation (co-IP) of Gtsf1-GFP from OSC nuclear extract using GFP-Trap. Control IPs were performed with OSCs transfected with a GFP-expressing vector. **b** Principle of dual-luciferase co-IP. A bait protein is fused to FLAG-tagged Firefly luciferase (FLAG-FFL), whereas potential interactors (preys) are fused to *Renilla* luciferase (RL). Following transient co-expression in S2R+ cells, the bait is immunoprecipitated with anti-FLAG antibodies and the co-IP efficiency is quantified by monitoring FFL and RL activities. **c** Graph shows the normalized co-IP efficiency (co-IP efficiency normalized to the RL-mCherry co-IP, this latter being used as a control of non-interacting prey) between the Gtsf1 bait and the indicated RL preys: Piwi, MEP-1, Mi-2, and Rpd3. Dots show values for $n = 4$ biologically independent samples, lines represent mean values ± SEM. **d** Western blot analysis of co-IP of endogenous Piwi and MEP-1 from Gtsf1-FH-containing OSC nuclear extract using anti-Piwi (left panel) and anti-MEP-1 (right panel) antibodies. Mouse and rabbit IgGs were used as control IPs for Piwi and MEP-1, respectively. The amount of input (In.) loaded relatively to IP is indicated as a percentage. Source data: **c**: Supplementary Data set 2; uncropped blot images are provided in Supplementary Data set 1.

anti-Piwi antibody to immunoprecipitate the endogenous Piwi from nuclear extracts of OSC cells expressing Gtsf1-FH. The known Piwi partner Gtsf1 (refs. [25,26]), as well as Mi-2, MEP-1, Rpd3, and p55 co-immunoprecipitated with Piwi in this experiment (Fig. 1d, left panel). The reciprocal immunoprecipitation of endogenous MEP-1 from the OSCs extract revealed the presence of Mi-2, Gtsf1, Piwi, Rpd3, and p55 in the immuno-precipitate (Fig. 1d, right panel). Specificity of the IPs was supported by the absence of Pc, a PRC1 Polycomb silencing complex component (Fig. 1d).

Altogether, these data provide evidence for physical interactions between Piwi, Gtsf1 and at least three components of chromatin modifying and remodeling co-repressor complexes.

**Rdp3, MEP-1, and Mi-2 transiently interact with piRNA machinery.** Rpd3, MEP-1, and Mi-2 are known to belong to several repressor complexes as follows: (1) a functional counter-part of the yeast SHREC, the *Drosophila* nucleosome remodeling and deacetylation (dNuRD) complex, is composed of a core complex containing MTA1-like, Rpd3, p55, and MBD-like to which Mi-2, CDK2AP1, SIMJ, and also MEP-1, may be less stably associated[33,40]; (2) a stable dMec also contains MEP-1 and Mi-2, but, unlike dNuRD, is devoid of Rpd3 (ref. [32]); (3) finally, several complexes contain the Rpd3/p55 module[41,42]. We attempted to determine whether Piwi co-elutes with any of these complexes, by performing Superose 6 gel filtration of nuclear extracts from OSC cells (Fig. 2a). Rpd3 and p55 were detected in several fractions representing a broad range of apparent molecular masses (100 to >2000 kDa) in agreement with the notion that these proteins are components of several distinct complexes. Very similar elution peaks (fractions 20–22) were observed for Mi-2 and MEP-1, matching the elution profile of the dMec stable complex pre-viously described in *Drosophila* Kc167 cell nuclear extracts[32]. A small proportion of both proteins was also present in a fraction of higher molecular weight where the dNuRD-specific MTA1-like protein peaked (fraction 18), in agreement with the typical dNuRD elution profile previously described for embryonic MTA1-like and Mi-2 (ref. [43]). However, neither of these two complexes co-eluted with an apparently stable Piwi-Gtsf1 com-plex, which reproducibly peaked in fractions of much lower apparent molecular mass (fraction 34). Thus, although physical interactions exist between Piwi, Gtsf1, Mi-2, MEP-1, Rpd3, and

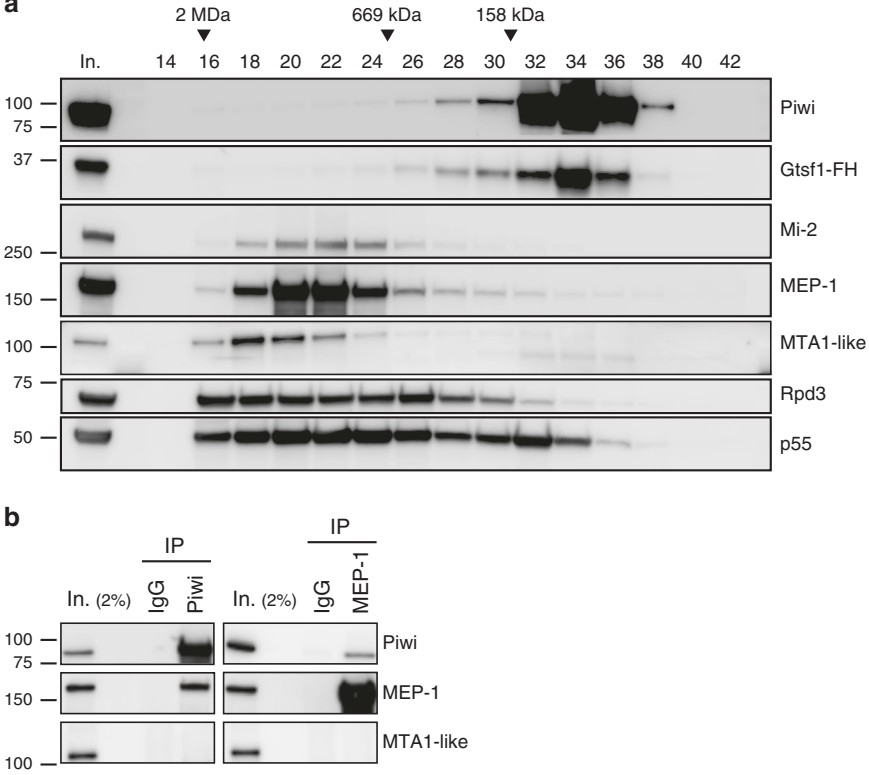

**Fig. 2 Rdp3, Mi-2, and MEP-1 are not associated with Piwi and Gtsf1 as a stable preformed subunit. a** Superose 6 gel filtration of nuclear extracts from OSCs. Fractions were analyzed by western blotting using the indicated antibodies. Fraction numbers and molecular mass standards are indicated on the top. Input (In.): 2% of extract loaded onto the column. **b** Western blot analysis of co-IP of endogenous Piwi and MEP-1 from OSC nuclear extract. Mouse and rabbit IgGs were used as control IPs for Piwi and MEP-1, respectively. The amount of inputs loaded relatively to IPs is 2%. Source data: uncropped blot images are provided in Supplementary Data set 1.

p55, they are either too transient or not abundant enough for a putative scaffold of these subunits to be isolated in our experimental conditions.

Next, as one of these complexes, the canonical dNuRD, may contain all three proteins together, at least in embryos[33], we examined whether it could be detected in Piwi and MEP-1 immunoprecipitation. To do so, we looked for the presence of the MTA1-like protein, a diagnostic dNuRD subunit, in co-IP experiments of endogenous Piwi and MEP-1 in OSC cells. Neither Piwi nor MEP-1 were able to pull down MTA1-like (Fig. 2b), suggesting that the bona fide dNuRD complex is not associated to Piwi. Taken together, these data suggest that Mi-2 and MEP-1 interact with Piwi not as components of dNuRD but as a dMec preformed module. Therefore, Rpd3 cannot be conveyed to the Piwi-dependent silencing machinery by the dNuRD complex, but, perhaps, as a component of another Rpd3/p55-containing co-repressor complex.

**Rpd3, Mi-2, and MEP-1 mediate TE transcriptional repression.** To study the impact of Rpd3, Mi-2, and MEP-1 on TE repression, we knocked them down (KD) by RNA interference (RNAi) in OSCs where TEs are repressed by a fully functional Piwi-piRNA pathway[44]. After validation of the RNAi efficiency (Supplementary Fig. 3a, b) and normalization with a siRNA GFP control (siGFP), we compared the mRNA-seq data between the Mi-2 or the Rpd3- and the Piwi-KD (Fig. 3a and Supplementary Fig. 3c). Most of the TEs derepressed upon Piwi depletion were also derepressed upon depletion of Mi-2 or Rpd3. For example, *mdg1* and *gypsy* were among the most highly derepressed TEs in Mi-2-, Rpd3-, and Piwi-KD experiments. Interestingly, similar to *piwi*, the *Mi-2* and *Rpd3* RNAi also affected the expression of the TE-

regulated gene *expanded* (*ex*)[20] (Fig. 3b). The derepression of three TE families (*Tabor*, *gypsy*, and *mdg1*), as well as *ex*, after Mi-2- and Rpd3-KD in OSCs was validated by reverse transcriptase-PCR quantification (RT-qPCR). We also found the same derepression effect of the MEP-1-KD on the tested TEs and on *ex* (Fig. 3c). In contrast, we confirmed genetically, that the dNuRD complex *per se* is not involved in TE repression. After MTA1-like, as well as MBD-like KD in OSC cells the RT-qPCR of steady-state mRNA levels of several piRNA targets did not reveal any mRNA increase for endogenous TEs nor for the *ex* gene (Supplementary Fig. 3d).

To analyze the function of MEP-1 and Mi-2 on TE repression in vivo, we depleted them in ovarian somatic tissues by expressing short hairpin RNAs under the control of the soma-specific *traffic-jam GAL4* driver. We used the *gypsy-lacZ* reporter to monitor silencing of a soma-specific TE (*gypsy*) in follicle cells[20]. X-Gal staining in MEP-1- and Mi-2-depleted ovaries indicated derepression of the *gypsy-lacZ* reporter (Fig. 4a, left panel). RT-qPCR confirmed the upregulation of the *gypsy-lacZ* mRNA (>30-fold increase) in these ovaries (Fig. 4b). It also showed an increase of the steady-state mRNA levels of the three tested endogenous TE families (*Tabor*, *gypsy*, and *mdg1*), which phenocopies the effects of the Piwi depletion on these TEs (Fig. 4b).

To prevent possible artefacts due to pleiotropic effects of the somatic KD on ovarian development, we restricted the KD to adult somatic ovarian cells, using the GAL80[ts] thermo-sensitive GAL4 inhibitor. In Piwi, MEP-1-, and Mi-2-partially depleted ovaries, showing a normal morphology, derepression of the *gypsy-lacZ* reporter in follicle cells was still observed (Fig. 4a, right panel), consistent with the increased levels of the soma-specific *gypsy* TE (and to a lesser extent of *Tabor* and *mdg1*) observed by

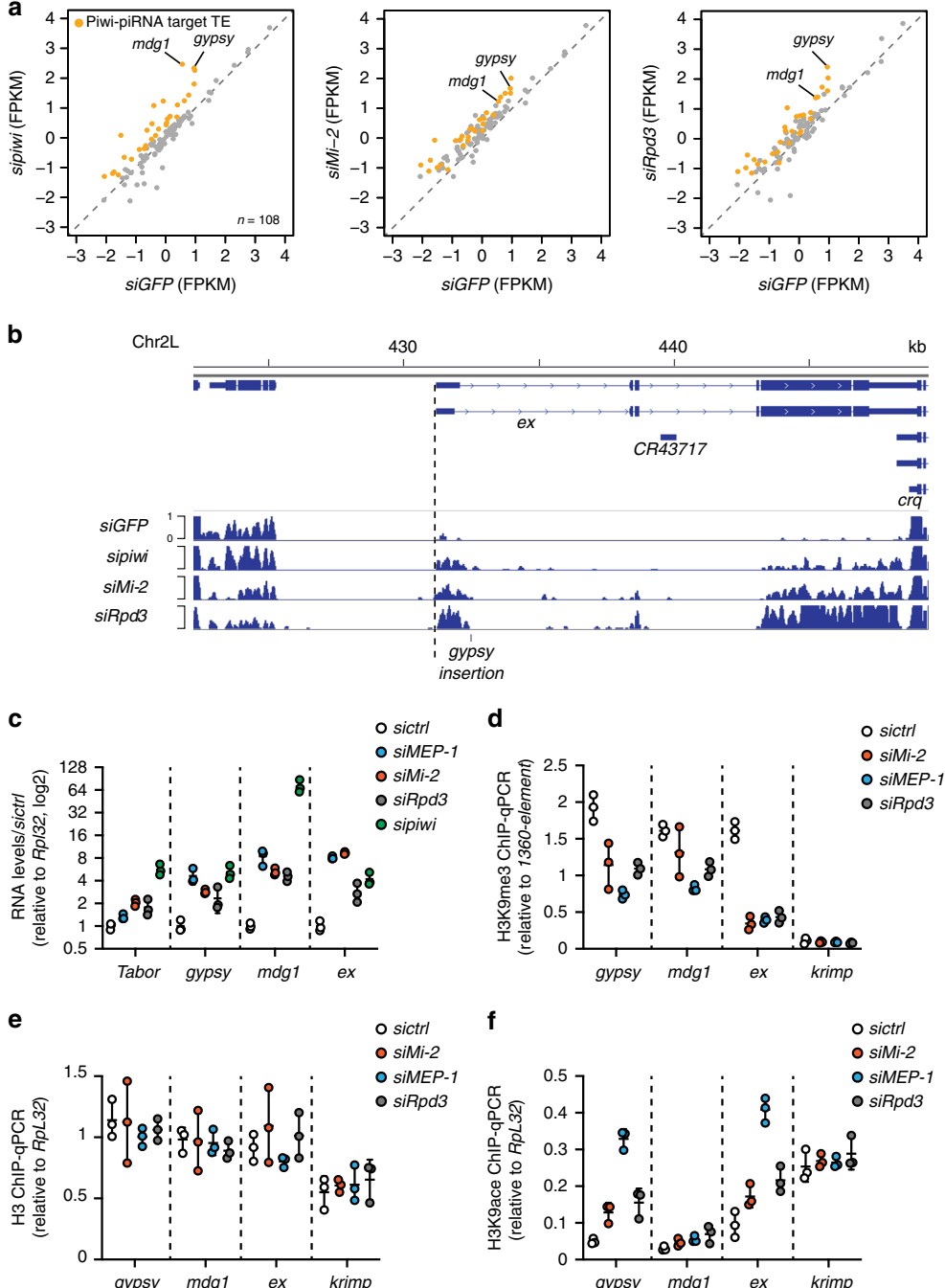

**Fig. 3 Rpd3, Mi-2, and MEP-1 are implicated in TE epigenetic silencing in OSCs. a** Scatterplots of RNA-seq reads in FPKM (fragments per kilobase of exon per million reads mapped) for 108 annotated *D. melanogaster* TEs in control knockdown (KD) (*siGFP*) vs. Piwi-KD (*sipiwi*) (left), Mi-2 KD (*siMi-2*) (middle), and Rpd3-KD (*siRpd3*) (right). TEs for which the expression level differed from control by more than two-fold in Piwi-KD (Piwi–piRNA-targeted TEs) are plotted in orange. Both *x*-axis and *y*-axis are a log10 scale. **b** A browser screenshot shows RNA-seq tracks upon GFP (*siGFP*), Piwi (*sipiwi*), Mi-2 (*siMi-2*), and Rpd3 (*siRpd3*) knockdown at the *expanded* (*ex*) locus. OSC-specific Gypsy insertion site is annotated; *ex* TSS is indicated with a dashed line. **c** RT-qPCR fold changes in steady-state RNA levels of three endogenous TEs (*Tabor*, *Gypsy*, and *mdg1*) and of the *expanded* (*ex*) gene upon MEP-1 (*siMEP-1*), Mi-2 (*siMi-2*), Rpd3 (*siRpd3*), or Piwi (*sipiwi*) knockdown using siRNAs. Dots show RNA values quantified relative to *Rpl32* and normalized to control knockdown for *n* = 3 biologically independent samples, lines represent mean values ± SD (log2). **d–f** H3K9me3 (**d**), H3 (**e**), or H3K9ace (**f**) quantified by ChIP-qPCR at *mdg1* and *Gypsy* TE genomic loci, as well as on the *expanded* (*ex*) and *krimper* (*krimp*) genes after knockdown using the indicated siRNA. Values relative to a positive control (*1360-element* for H3K9me3 or *RpL32* for H3 and H3K9ace) were normalized to input (mean ± SD from *n* = 3 independent biological replicates). Source data for **c**–**f**: Supplementary Data set 3.

RT-qPCR in total ovarian extracts (Fig. 4c). After RNAi against Rpd3, the LacZ expression was irregular from one ovariole to another, from weak to null, preventing us from concluding on the effect of Rpd3 depletion in vivo. The discrepancy between the effects of Rpd3 depletion in vivo and ex vivo could be due to a possible low efficiency of the RNAi in flies. In summary, a functional role in TE repression was demonstrated ex vivo for Rpd3 and both ex vivo and in vivo for Mi-2 and MEP-1.

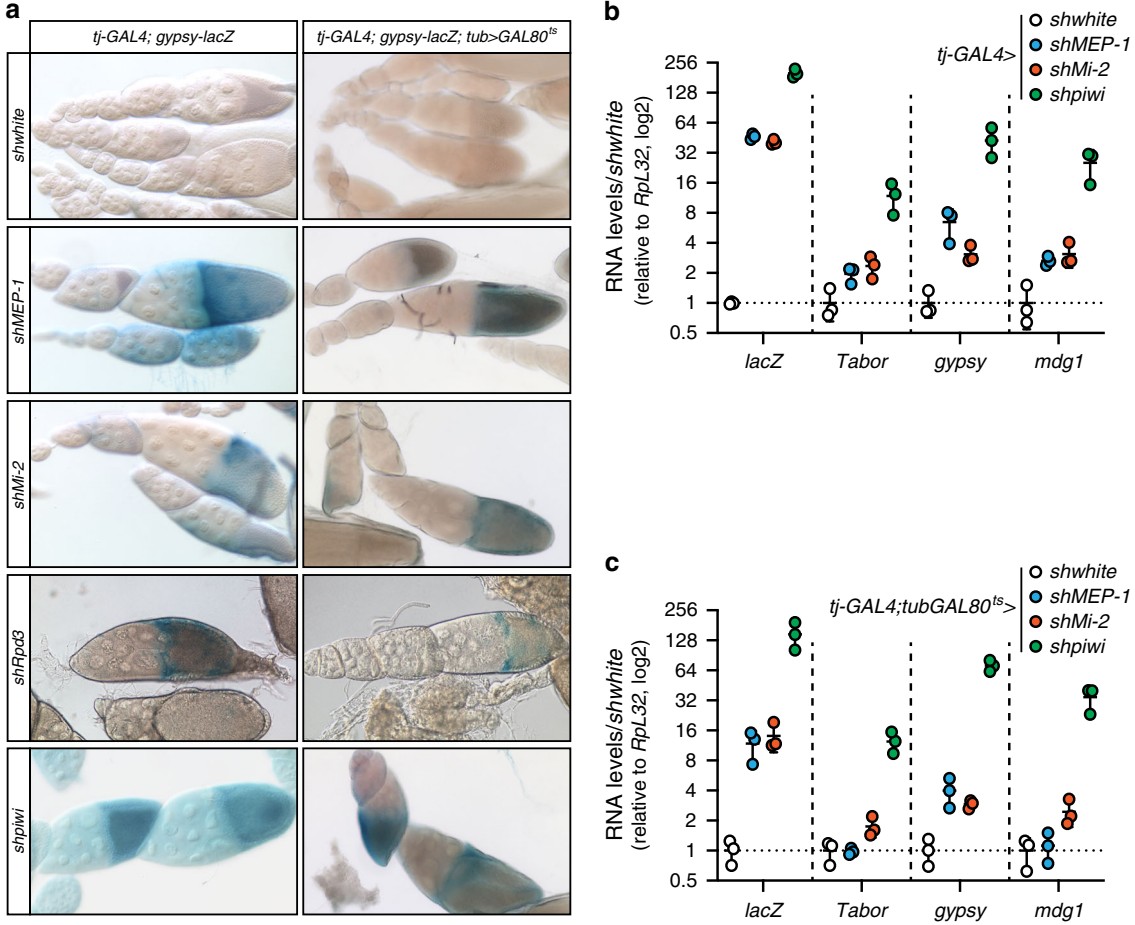

**Fig. 4 The ovarian somatic depletion of MEP-1 or Mi-2 results in TE and TE reporter derepression in vivo. a** X-Gal staining of egg chambers that contain the somatic TE reporter (*Gypsy-lacZ*) in which the indicated genes were knocked down by constitutive (left panel) or conditional (right panel) somatic RNAi. Conditional somatic knockdown was restricted to adult flies. The control was a shRNA against the *white* gene (*shwhite*). **b, c** Fold change (log2) in the steady-state RNA levels of the *Gypsy-lacZ* reporter (lacZ) and of three endogenous TEs in ovaries after constitutive (*tj-GAL4 > sh*) (**b**) or conditional (*t-jGAL4;tubGAL80[ts] > sh*) (**c**) somatic knockdown, using the indicated RNAi. RNA levels were quantified relative to *RpL32* levels and normalized to control knockdown (*shwhite*). Lines represent means ± SD for *n* = 3 biologically independent samples. Source data for **b, c**: Supplementary Data set 4.

We then asked whether the derepression in OSCs correlates with a decrease of the H3K9me3 repressive mark on endogenous TEs and on the *ex* gene. Indeed, it was previously reported that Piwi-dependent repression of *ex* (via the targeting of an OSC-specific TE insertion in the first intron of this gene) correlates with H3K9me3 spreading up to the *ex* transcription start site[20]. Each of the MEP-1, Mi-2, and Rpd3-KD correlated with a loss of H3K9me3 on the chromatin of *ex* and to a lesser extent of the two tested TE families (*gypsy* and *mdg1*) (Fig. 3d). This decrease in H3K9me was not due to a loss of nucleosomes, as chromatin IP (ChIP)-qPCR did not show any change in the histone H3 occupancy levels at these loci (Fig. 3e). We also observed an increase of the H3K9ace active mark on *gypsy* and *ex*, which appears very pronounced under MEP-1 KD (Fig. 3f).

As MEP-1, Mi-2, and Rpd3 are involved in chromatin-mediated regulation of gene expression, they could also be required for piRNA biogenesis, instead of participating in the mechanism of TE repression. To choose between these hypotheses, we first performed RT-qPCR of piRNA precursor transcripts and of mature piRNAs in OSCs following silencing of either *MEP-1*, *Mi-2*, or *Rpd3*, as well as *piwi* for a positive control. For none of the three depleted proteins, we have observed any decrease in steady-state piRNA precursors nor mature piRNAs levels (Supplementary Fig. 4a, b). Moreover the piRNA loading-

dependent nuclear Piwi immunolocalization[44–46] was unchanged in MEP-1-depleted OSCs (Supplementary Fig. 4c).

Altogether, these observations suggest that the functions of MEP-1, Mi-2, and Rpd3 in the transcriptional silencing of TEs and of their flanking genes do not involve the piRNA production pathway but rather the repression mechanism itself.

**TE repression by Rdp3, MEP-1, and Mi-2 is piRNA dependent.** To determine whether MEP-1, Mi-2, and Rpd3-mediated TE repression is piRNA-dependent, we took advantage of a previously published assay that allows the exclusive analysis of piRNA-mediated repression on an endogenous gene, *krimper* (*krimp*), in OSCs (Fig. 5a). This assay is based on the ability of a transfected plasmid to produce artificial Piwi-associated piRNAs (apiRNAs) against this gene[47]. Production of these apiRNAs in OSCs led to an almost two-fold de novo repression of *krimp* expression (Fig. 5b). This apiRNA-mediated repression of *krimp* was significantly reduced upon MEP-1, Mi-2, or Rdp3 KD (Fig. 5b). These data show that these three proteins are as important as Piwi for an efficient apiRNA-mediated repression in this system. To rule out an apiRNA-independent repression of *krimp* reporter by these proteins, we checked *krimp* mRNA levels after MEP-1, Mi-2 and Rpd3 depletion in the absence of

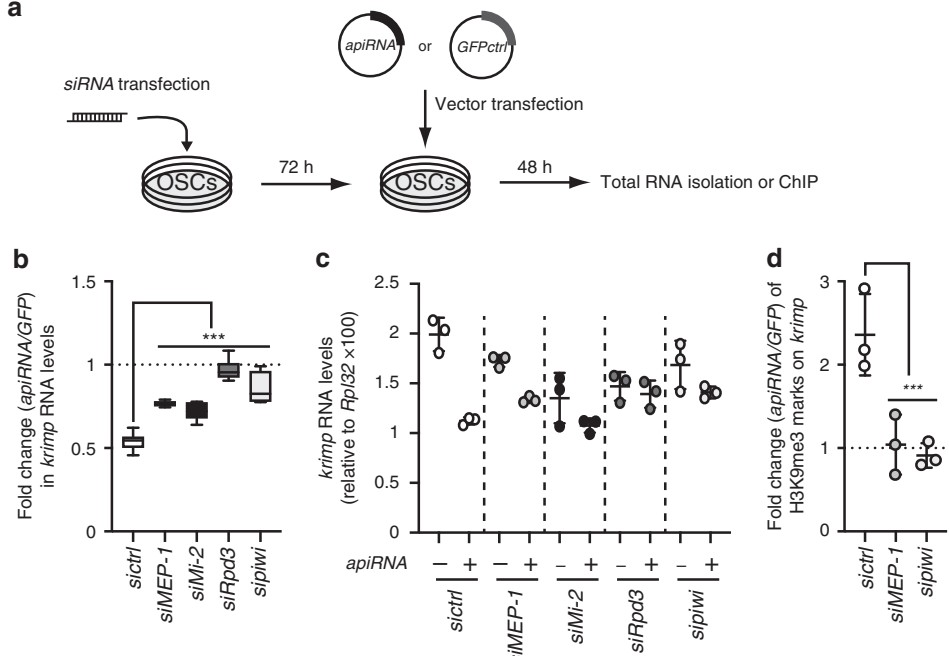

**Fig. 5 MEP-1, Mi-2, and Rpd3 involvement in piRNA-dependent repression. a** Schematic representation of the *krimper* (*krimp*) piRNA-mediated repression assay after siRNA-mediated depletion of MEP-1, Mi-2, Rpd3, or Piwi in OSCs. Transfection of the apiRNA vector allows the production of antisense artificial piRNAs against *krimp*. Transfection of the GFPctrl vector is used as a piRNA-less negative control. **b** Fold change (relative to GFPctrl) in *krimp* RNA levels caused by apiRNA in OSCs previously transfected with the indicated siRNAs. RNA levels were quantified relative to *RpL32*. Box plots display median (line), first and third quartiles (box), and highest/lowest value within 1.5× interquartile range (whiskers) for $n = 6$ values calculated over three independent samples. The siRNA effect was tested using the ANOVA test and differences using the pairwise t-test. P-values were calculated using sample data that displayed normal distribution (tested with the Shapiro–Wilkinson test). Variance homogeneity was tested with the Levene's test and then the two-tailed Student's *t*-test was used. ***P-value < 0.05 when each siRNA experiment is compared to the *sictrl* experiment. **c** Quantification of *krimp* RNA levels relative to *RpL32* in OSCs transfected by the indicated siRNAs with (+) or without (−) production of apiRNAs against *krimp*. Note that the cells that were not transfected with the apiRNA vector were transfected with GFPctrl vector, instead. Means ± SD from $n = 3$ biologically independent samples are represented. **d** Fold change (relative to GFPctrl) of H3K9me3 marks on *krimp* quantified by ChIP-qPCR. OSCs were transfected with the indicated siRNAs and either the apiRNA or the GFPctrl vector. H3K9me3 quantification was normalized to a positive control (1360-element) and to input. Means ± SD from $n = 3$ biologically independent samples are represented. Statistical test was performed as in **b**. ***P-value < 0.05 compared with *sictrl*. Source data for **b**–**d**: Supplementary Data set 5 and Supplementary Table 2 for **b**, **c**.

apiRNAs. We observed that the knockdown of these proteins caused a slight decrease of *krimp* mRNA levels in the absence of apiRNAs but not an increase, which would be expected in case of apiRNAs-independent repression (Fig. 5c). Moreover, the impairment of the apiRNA-mediated *krimp* repression by MEP-1 and Piwi-KD, was correlated with a loss of apiRNA-mediated H3K9me3 deposition at this locus (Fig. 5d). Altogether, these data indicate that MEP-1, Mi-2, and Rpd3 may be considered not only as physical but also as functional Piwi partners involved in piRNA-dependent transcriptional repression.

**MEP-1, a putative scaffolder of the Piwi silencing machinery.** Our data suggest that MEP-1 and Mi-2 form a stable dMec subunit of the Piwi-dependent TE repression machinery. In *Drosophila*, dMec is known as a SUMO-dependent co-repressor complex[17] that is guided to its DNA target via binding to a SUMOylated transcription factor. The deposition of the repressive chromatin marks via SUMOylation seems to be a general mechanism that contributes to the recruitment of chromatin-associated proteins, thereby promoting target gene silencing[48]. For instance, it has been proposed that the interaction of the piRNA-loaded Piwi protein with the SUMO E3 ligase Su(var)2-10 allows the recruitment of the histone methyltransferase complex, SetDB1-Windei, resulting in TE transcriptional repression[24]. As

Su(var)2-10 SUMOylates itself, we aimed to determine if Su(var)2-10 physically interacts with the MEP-1/Mi-2 module. To this end, we overexpressed a FFL Su(var)2-10 protein (FLAG-FFL-Su(var)2-10) in OSC cells and performed IPs using an anti-MEP-1 antibody. We found that MEP-1 interacts with Su(var)2-10 and SetDB1 in OSC cells (Fig. 6a, left panel). For the Su(var)2-10 reciprocal IP, as there is already a FLAG tag in our cells (Gtsf1-FH), we overexpressed a GFP-tagged Su(var)2-10 protein (Su(var)2-10-GFP) to avoid interference. We found that Su(var)2-10 interacts not only with SetDB1 and Piwi in OSC cells, but also with MEP-1 and Mi-2 (Fig. 6a, right panel). Our data are consistent with a model in which dMec is part of previously described SUMO-dependent chromatin-associated transcriptional repression machinery.

In *Caenorhabditis elegans*, MEP-1 interacts with the SUMOylated LIN-1 transcription factor to induce the transcriptional repression. To further study how MEP-1 interacts with Su(var)2-10, we performed LUMIER assays in which the FFL Su(var)2-10 protein was used as a bait to quantify its interaction with preys fused to RL in S2 cells. We first confirmed that Su(var)2-10 can directly or indirectly interact with SetDB1 and we revealed also an interaction with MEP-1 and Mi-2 (Fig. 6b). The binding of Su(var)2-10 to MEP-1 is likely mediated by the C terminus of MEP-1, as deletion of the last 345 amino acids in the MEP-1 sequence (MEP-1ΔCt) reduced the interaction (Fig. 6b). By contrast, this

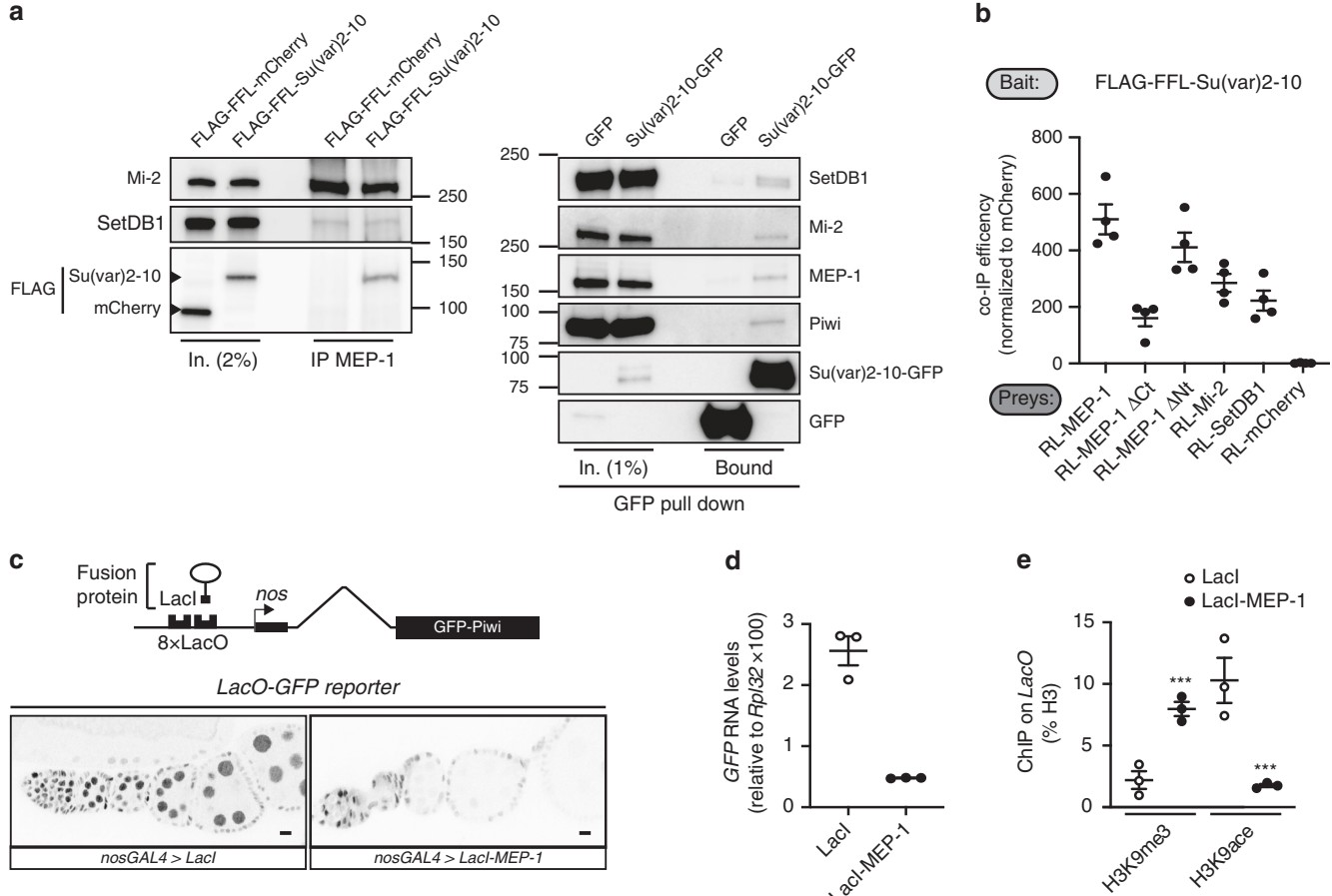

**Fig. 6 MEP-1 interacts with Su(var)2-10 and induces transcriptional silencing upon recruitment to DNA. a** Western blot analysis of co-IP of (left panel): MEP-1 in OSC nuclear extract expressing either FLAG-FFL-Su(var)2-10 or FLAG-FFL-mCherry as a negative control or of (right panel): GFP-Trap in OSC nuclear extract expressing either Su(var)2-10-GFP or GFP alone. **b** Graph shows the normalized co-IP efficiency of several preys (MEP-1, MEP-1ΔCt, MEP-1ΔNt, Mi-2, and the negative mCherry control) with the FLAG-FFL-Su(var)2-10 bait. Dots show values of $n = 4$ biologically independent samples, lines represent mean ± SEM. **c** Targeting MEP-1 to reporter DNA induces silencing in vivo. Schematic representation of the *8xlacO-GFP* reporter. The lower panels show the GFP fluorescence signal of *LacO-GFP* reporter in egg chambers that express the LacI (left) or LacI-MEP-1 fusion proteins (right) under the control of the germline-specific *nos-GAL4* driver. Scale bars, 10 μm. **d** Dots showing quantification of *eGFP* RNA levels in ovaries that express LacI or LacI-MEP-1 under the control of the germline-specific *nos-GAL4* driver. RNA levels are relative to *RpL32* level. Mean values ± SEM of $n = 3$ biologically independent samples are represented. **e** Comparison of H3K9me3 and H3K9ace amounts on the *LacO-GFP* reporter, expressed in percentage of immunoprecipitated H3, in ovaries that express LacI or LacI-MEP-1 fusion proteins. *P*-values were calculated using sample data that displayed normal distribution (tested with the Shapiro–Wilkinson test). Variance homogeneity was tested with the Levene's test and then the two-tailed Student's *t*-test was used. \*\*\**P*-value < 0.05. Mean values ± SEM of $n = 3$ biologically independent samples are represented. Source data for **b**, **d**, **e**: Supplementary Data set 6 and Supplementary Table 2 for **e**; uncropped blot images are provided in Supplementary Data set 1.

interaction was not strongly perturbed after removal of the first 680 MEP-1 amino acids (MEP-1ΔNt) that contain the binding site of Mi-2 (ref. [32]). Altogether, these data suggest that MEP-1 is not passively guided to the silencing machinery as an obligatory Mi-2 partner but might use its C terminus to interact with other components of the silencing machinery.

To determine whether MEP-1-mediated targeting to DNA would be sufficient to silence the targeted locus, we tethered a LacI-MEP-1 fusion protein to the previously described LacO-GFP transgene[21] (Fig. 6c). In ovarian germ cells, MEP-1 tethering induced reporter silencing (Fig. 6c–d) that correlated with increased H3K9me3 and decreased H3K9ace levels (Fig. 6e). These effects are consistent with the MEP-1 interactome, including SetDB1 (Fig. 6a) and Rpd3 (Fig. 1c), respectively. Altogether, these data support a model in which MEP-1 is part of a transcriptional repression machinery involving both chromatin modifier and remodeler enzymes essential for the piRNA-dependent heterochromatin formation that enforce TE repression.

## Discussion

In most eukaryotes, the transcriptional silencing of TEs is associated with H3K9me3. In *Drosophila* ovaries, the piRNAs loaded on the Piwi protein are involved in the targeting of TE nascent transcripts, by sequence complementarity, for the deposition of this heterochromatic repressive mark[19,20,49] by the SetDB1 methyltransferase[21–23]. The molecular mechanism of this heterochromatin formation is still not fully understood.

Here we report that a chromatin eraser, Rpd3, and a nucleosome remodeler, Mi-2, in complex with its partner MEP-1, are required for the Piwi-piRNA-dependent TE silencing in OSC cells. The depletion of Mi-2 and MEP-1 also results in TE transcriptional derepression in fly ovaries (Figs. 3, 4). Moreover, they physically interact together and with Piwi in OSC cells (Fig. 1). Even though Rpd3, MEP-1, and Mi-2 do not seem to form a stable multiprotein complex associated with Piwi, we propose that at least three preformed subunits Piwi/Gtsf1, Mi-2/MEP-1, and Rpd3/p55 transiently interact to build up the Piwi-dependent TE

silencing machinery. We reveal that this TE transcriptional repression correlates with an increased H3K9 trimethylation and also with a decrease in H3K9 acetylation (Fig. 3d, f) that is consistent with the Rpd3 histone deacetylase being a Piwi interactor (Fig. 1). Our data suggest also that, in S2 and OSC cells, MEP-1 interacts with both SetDB1 and the SUMO ligase Su(var)2-10, another Piwi interactor (Fig. 6a). This interaction is not simply mediated by its Mi-2 partner, since it is reduced when using a MEP-1ΔCt mutant, which is still able to bind Mi-2.

To our knowledge, dNuRD is the only complex that can combine chromatin remodeling with histone deacetylase activities in *Drosophila*. This multiprotein complex is composed of a core complex containing Rpd3, p55, MTA1-like and MBD-like to which Mi-2, and also MEP-1, may be less stably associated[40]. However, we found that neither Piwi nor MEP-1 physically interact with MTA1-like (Fig. 2b), and that neither MTA1-like nor MBD-like are genetically involved in TE silencing (Supplementary Fig. 3d). Therefore, we conclude that, instead of the classical dNuRD complex, Mi-2 likely exerts its TE silencing function in OSC cells as part of a smaller Mi-2/MEP-1 module which, in size-exclusion chromatography, behaved very much like the dMec complex already described in *Drosophila* Kc167 cells[32] (Fig. 2a).

Indeed, in Kc167 cells, both Mi-2 and MEP-1 are known to strongly interact together, to bind to SUMO and to be recruited to some promoters by SUMOylated transcription factors, thereby inducing the transcriptional gene silencing in a HDAC-independent manner[17,48]. Of note, a SUMOylated Piwi interactor, Su(var)2-10, has recently been proposed to link piRNA-guided target recognition to chromatin silencing[24]. Interestingly, MEP-1 and Mi-2 interact with Su(var)2-10, the interaction of Su(var)2-10 with MEP-1 seems to be dependent of the C terminus of MEP-1 (Fig. 6a, b). However, unlike dMec-dependent gene repression in Kc167 cells, which does not recruit any HDAC to the repressed genes, in OSC cells, the transcriptional TE silencing involves the histone deacetylase Rpd3. This observation suggests that a more sophisticated silencing mechanism is involved in TE than in gene repression. A thorough characterization of the Rpd3-containing module of the TE silencing machinery is still needed.

A tentative model of the assembly of the Piwi-dependent silencing machinery will also have to explain why we could only visualize sub-stoichiometric interactions between all three sub-units of this putative repressing complex (Fig. 1d). A first trivial possibility is that such interactions were destabilized by our extraction procedure. A second explanation is that only a small fraction of each protein pool is involved in the silencing machinery. For example, most Piwi proteins are likely busy with scanning the transcriptome, guided by their associated piRNAs, in search of a cognate target. Recruitment of Piwi partners to mediate transcriptional repression seems to only occur when Piwi has found the nascent RNA that is able to hybridize with its bound piRNA[21]. A third hypothesis is that the interactions between the different partners occur only transiently in the silencing machinery. Indeed, the targeted locus, in the Piwi-dependent transcriptional repression, is chosen via a transient nascent RNA that may not stay long in close proximity to the DNA. This transcription-dependent repression is different from the recruitment to chromatin of co-repressor complexes by their stable association with DNA-bound transcription factors.

In *S. pombe* too, heterochromatin formation may rely on the cooperation between a RNAi-mediated targeting system and a chromatin silencing machinery. Two of the chromatin factors, a histone deacetylase (Clr3) and a nucleosome remodeling ATPase (Mit1) are integral components of the SHREC complex[15,50]. SHREC is not a preformed complex, but it seems to result from the scaffolding of both these autonomous sub-entities after their

independent recruitment at the target. Similarly, our data suggest that, in *Drosophila*, (1) two functional homologs of these chromatin effectors, respectively Rpd3 and Mi-2, are also required for piRNA-mediated transcriptional silencing; (2) these proteins are not associated in a preformed complex either. These similarities provide further evidence for the conservation of general mechanisms of small RNA-mediated heterochromatin formation, whatever the small RNA involved (siRNAs in yeast or piRNAs in *Drosophila*).

## Methods

**OSC cell culture and transfection**. Wild-type OSCs obtained from fGS/OSS[44] and genome-edited OSCs expressing tagged Gtsf1 were cultured in Shields and Sang M3 Insect medium (USBiological) supplemented with 10% fetal bovine serum, 10% fly extract, 0.6 mg/ml L-glutathione reduced, and 0.01 mg/ml insulin (i.e., complete medium) at 24 °C (refs. [31,44]). For OSCs expressing tagged Gtsf1, the medium was supplemented with 25 μg/ml blasticidin (InvivoGen). Cells ($3–5 \times 10^6$) were transfected with 200 pmol siRNA duplex in 100 μl Mirus Ingenio® solution using the Amaxa Nucleofector II (program setting T-029). Transfected cells were plated in 6-well culture dishes with 1.4 ml complete medium and incubated for 24–96 h (depending on siRNA) at 24 °C, collected, and used for immunoprecipitation, ChIP, or RNA quantification.

For the *krimper* experiments, siRNA-transfected OSC cells were grown in complete medium for 72 h and then transfected with 4 μg *krimp*-targeting artificial piRNA expression vector or GFP control using the Xfect transfection reagent (TaKaRa)[47]. Cells were incubated at 24 °C for 48 h and then collected.

For the transient overexpression of C-terminally GFP-tagged Gtsf1 and Su(var)2-10, cDNAs were cloned in the pUWG (DGRC:1284) vector and transfection was performed using Xfect Transfection Reagent. Empty vector was used as control.

For CRISPR/Cas9 DmGtsf1 genome editing: see Supplementary Information.

GST-pull down assays and methods for Protein Sequence Analysis by liquid chromatography–tandem mass spectrometry are described in Supplementary Information.

**Immunoprecipitation and western blotting of OSC nuclear extracts**. IPs were performed using the Nuclear Complex Co-IP Kit (Active Motif) under "stringent" conditions following the manufacturer's instructions. For each IP, 10 μl of anti-Piwi mouse monoclonal antibody (Santa Cruz), 2.5 μl of anti-MEP-1 rabbit polyclonal antibody, or 2.5 μg equivalent of control IgG were added to 500 μg proteins (adjusted to 500 μl in IP buffer) and incubated overnight at 4 °C with rotation. Fifty microliters of Dynabeads protein G (Invitrogen) were then added and incubated at 4 °C for 2 h with rotation. The beads were then washed 6 times 5 min with 500 μl IP buffer and bound proteins were eluted in 30 μl of 2× Laemmli buffer (BioRad) at 95 °C. Western blotting was done following standard protocols. Proteins were separated using 4–15% TGX stain free polyacrylamide gels (BioRad) and transferred to 0.45 μm poly-vinylidene difluoride membranes (Millipore). The membranes were blocked in 5% skimmed milk in TBS supplemented with 0.1% Tween 20 (TBST) for 1 h and incubated with primary antibodies overnight at 4 °C. After 3 washes in TBST, membranes were incubated with horseradish peroxidase-conjugated secondary antibodies for 1 h at room temperature (RT), followed by three washes in TBST.

**Gel filtration**. For the fractionation of nuclear OSC extracts, cells were collected, washed in ice-cold phosphate-buffered saline (PBS), and resuspended in five volumes of hypotonic buffer (10 mM Hepes pH 7.8, 1.5 mM MgCl₂, 10 mM KCl, 1 mM dithiothreitol (DTT), 0.2 mM phenylmethylsulfonyl fluoride (PMSF) and protease inhibitor cocktail (Roche)). After incubation on ice for 15 min, cells were centrifuged at 16,000 × g at 4 °C for 10 min, and the pellet was resuspended in two volumes of salt buffer (20 mM Hepes pH 7.8, 1.5 mM MgCl₂, 300 mM NaCl, 1 mM DTT, 0.2 mM PMSF and protease inhibitor cocktail). The suspension was rotated for 30 min at 4 °C and then centrifuged at 100,000 × g for 30 min at 4 °C on a Beckman Optima TLX-120 Ultracentrifuge. The supernatant (nuclear extract) was collected and the protein concentration was determined using the Bradford assay (BioRad). The nuclear extract was applied to a Superose 6 gel filtration column (HR 10/30 GE Healthcare) using a 500 μl sample loading loop on an Äkta purifier system (GE Healthcare) and resolved in 10 mM Hepes pH 7.8, 1.5 mM MgCl₂, 200 mM NaCl, and 0.2 mM DTT. Fractions (0.5 ml) were collected and precipitated with 20% (final concentration) trichloroacetic acid (Sigma) before western blot analysis. Calibration of the Superose 6 column was performed with protein standards of known molecular weights using the Gel Filtration Calibration Kit (GE Healthcare).

**Immunocytochemistry**. OSCs were plated on concanavalin A-coated coverslips, fixed with 4% paraformaldehyde in PBS (RT, 10 min), washed three times with PBS, permeabilized with 0.1% Triton X-100 in PBS (15 min), and blocked with 2% bovine serum albumin, 0.02% Tween 20 in PBS. Primary antibodies were against Piwi G-1 (1:500, mouse, Santa Cruz sc-390946) and MEP-1 (1:500, Rabbit, A. Brehm). Nuclei were stained with DAPI (Sigma). Images were captured with a Zeiss Axioimager Apotome microscope.

**Quantitative ChIP**. ChIP of ovaries and of OSCs, see Supplementary Information.

**X-Gal staining**. Ovaries from 5-day-old flies were dissected in PBS, kept on ice, fixed in 0.2% glutaraldehyde/2% formaldehyde/PBS at RT for 5 min, washed 3 times with PBS and incubated in staining solution (1× PBS pH 7.5, 1 mM MgCl$_2$, 4 mM potassium ferricyanide, 4 mM potassium ferrocyanide, 1% Triton, 2.7 mg/ml X-Gal) at 37 °C.

**RNA extraction and quantitative RT-PCR**. Total RNAs were isolated from ovaries or OSCs with Trizol, combined with Direct-zol™ RNA Miniprep Plus (Zymo research). Five hundred nanograms of total RNA was reverse transcribed using random primers and SuperScript III reverse transcriptase (Invitrogen). Quantitative PCR was performed using the LightCycler 480 SYBR Green I Master system (Roche). Each experiment was performed in biological triplicates and technical duplicates. Primer sequences used for RT-qPCR are listed in Supplementary Table 1.

**miRNA and piRNA quantification**. Total RNA was extracted with TRI Reagent® (Molecular Research Centre, Inc.) following the manufacturer's recommendations. Total RNA was used for cDNA synthesis and polyadenylation[51,52]: 100 ng of RNA in a final volume of 10 μl including 1 μl of 10× poly(A) polymerase buffer, 0.1 mM of ATP, 1 μM of RT-primer, 0.1 mM of each dNTPs, 100 units of Superscript II (Invitrogen) reverse transcriptase, and 1 unit of *E. coli* Poly(A) polymerase (M0276S, NEB) was incubated at 37 °C for 10 min, transferred at 42 °C for 50 min, and then at 70 °C for 10 min for enzyme inactivation. The sequence of the RT-primer was 5′-CAGGTCCAGT15VN-3′.

**mRNA-seq**. For the RNAi, trypsinized OSCs (3 × 10⁶ cells) were suspended in 20 μl of Solution SF of the Cell Line Nucleofector Kit SF (Amaxa Biosystems) together with 200 pmol of siRNA duplex. Transfection was conducted in a 96-well electroporation plate using a Nucleofector device 96-well Shuttle (Amaxa Biosystems). The transfected cells were transferred to fresh OSC medium and incubated at 26 °C for 4 days. Total RNAs were isolated using ISOGENII (Nippon Gene) according to the manufacturer's instructions. Purification of poly-A RNAs, preparation of libraries, sequencing by HiSeq2500 (Illumina), and adapter-trimming were done by BGI. The reads were mapped to *D. melanogaster* Release 6 (dm6) genome assembly and transcriptome (gene and transposon) by bowtie2 (ver. 2.2.4) using default parameters. The datasets were downloaded through piPipes[53]. Read counts corresponding to each genomic and genic position were obtained by generating bedgraph files from BAM files (binary version of SAM files) using the BEDTools genomecov. All samples were normalized to have the equivalent of reads per million using the "-scale" option. Fragments per kilobase of exon per million mapped fragments of each gene and transposon were manually calculated. For data visualization, R-packages implemented in R 3.2.1 was used.

**Dual-luciferase co-immunoprecipitation in S2R+cells**. cDNAs encoding Su (var)2-10, Gtsf1, and SetDB1 were obtained by RT-PCR from total RNA extract from OSC cells (see primers in Supplementary Table 1) and were cloned in vector pENTR/D-Topo, using NEBuilder HiFi DNA Assembly Cloning Kit (E5520S, NEB). The pDON223-Rpd3 was obtained from (DNASU Plasmid Repository: DmCD00768356), pENTR-TOPO-Mi-2 was a gift from A. Brehm, MEP-1 cDNA from (BDGP_cDNA: RE60032), and Piwi cDNA was a gift from M.C. Siomi. The mCherry cDNA used as a negative control in IP experiments was from Addgene plasmid #128744. To obtain luciferase-tagged proteins, these cDNAs were recombined using the gateway system into destination vectors pAct-Flag-Firefly-RfA, pAct-HA-Renilla-RfA[39] with LR clonase II (11791020, Invitrogen). The details protocol is presented in Supplementary information.

**Drosophila husbandry, strains and LacI-MEP-1 transgenic flies**. See Supplementary Information and Supplementary Table 3.

**Reporting summary**. Further information on research design is available in the Nature Research Reporting Summary linked to this article.

## Data availability
Data are accessible in the NCBI Gene Expression Omnibus (GEO; https://www.ncbi.nlm.nih.gov/geo/) under the accession number GSE141237. The mass spectrometry data have been deposited to the ProteomeXchange Consortium via PRIDE[54] partner repository with the data set identifier PXD018749 and 10.6019/PXD018749. Source data for Fig. 1a, d, 2, and 6a, and Supplementary Fig. 2b, c and 3a are available in Supplementary Data set 1, for Fig. 1c in Supplementary Data set 2, for Fig. 2c–f in Supplementary Data set 3, for Fig. 4b, c in Supplementary Data set 4, for Fig. 5b–d in Supplementary Data set 5, for Fig. 6b, d, e in Supplementary Data set 6, for Supplementary Fig. 3b, d in Supplementary Data set 7 and for Supplementary Fig. 4a, b in Supplementary Data set 8. Source data are provided with this paper.

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

## Acknowledgements

We thank Masaru Ariura for helping with the RNA-seq experiment. We thank the VDRC and Bloomington stock collections for transgenic RNAi lines, *Drosophila* Genomics Resource Center, supported by NIH grant 2P40OD010949 for vectors. We thank C. Dargemont for discussion, comments, and for critically reading the manuscript. We thank K. Förstemann, J. Brennecke, A. Brehm, J. Kadonaga, R. Pillai, Y. Yu, and W. Theurkauf for providing antibodies, vectors, and transgenic flies, and H. Seitz for statistics and discussion. We also thank the *Drosophila* facility, BioCampus Montpellier, CNRS, INSERM, Université de Montpellier. This work was supported in part by the Fondation pour la Recherche Médicale, grant number "DEQ20180339167" to S.C., the Fondation ARC "PJA20151203231," and the CNRS. K.S. is supported by JSPS KAKENHI grant 20K06596. M.C.S. is supported by MEXT grant 19H05466.

## Author contributions

B.M. designed and performed most of the experiments and prepared figures and tables. S.N. performed the immunoprecipitation and the Superose 6 gel filtration experiments. C.V.C. did the GST-pulldown experiment and analyzed the mass-spec data. C.J. did LUMIER assays. F.J. and E.B. gave advice for LUMIER assays. K.S. and M.C.S. did the mRNA-seq. A.P. performed *Drosophila* genetic crosses. A.P. and S.C. conceived the project and wrote the manuscript.

## Competing interests

The authors declare no competing interests.
