## [Peer Review File · Nature Communications]

Reviewers' comments:

Reviewer #1 (Remarks to the Author):

Here the authors present the results showing the roles of NuRD subunits in TE repression. It is pretended that the specific piRITS nuclear complex has been isolated using IP with Gtsf-FLAG-HA expression. Mass spec analysis of GST-GTSF IP shows significant enrichment of several putative components of this complex, including two NuRD subunits as well as a lot of some other associated proteins (Fig S1b). Re-immunoprecipitation of this IP using Ab against MEP-1 confirms the presence of putative complex components, but re-mass spec analysis of re-IP is needed to show further complex purification as well as some hints to reveal the stoichiometric presence of putative RITS components. Possibly CL approach and MS analysis would be desirable to analyze re-precipitate. The author's attempts in favor to detect the existence of this complex using glycerol centrifugation were unsuccessful (Fig.1c). So today it is possible to discuss only the transient co-association of the studied components. The sentence (lines 93-95) related to the complex structure discussion looks inappropriate. Fig.4 demonstrates only an ability of HDAC recruitment by MEP-1 as an itself peculiar NURD functional property. Fig.6S presents Egg/SetDB as a component of pi-RITS complex in the absence of a required direct experiment demonstrating this interaction. Position of Egg peptide is not even represented in Fig.S2b. Thus the author's statement in the abstract that they «present the first biochemical and genetic characterization of the Drosophila piRITS complex» is exaggerated.

The distinct part of this paper concerns the role of NuRD subunits in TE repression, but the relation of these results to the existence of postulated pi-RITS complex remains under question. The observation that piRNA induced krimp repression is similarly dependent upon Piwi, MEP-1 and Mi-2 KDs is an indication but not the evidence of postulated complex activity. The same comment concerns the observation of MEP-1 recruitment of HDAC. Thus, these observations themselves are of some interest, but they cannot serve as cardinal proof of the existence of a complex.

Minor comments

Fog 1a Absence of designations above the strips

It is unclear why fraction 4 content (Fig.1c) is selected to evaluate whether any component will be related to a putative complex.

Fig.2a. The results related to HetA are not clearly presented, picture replacement required

Reviewer #2 (Remarks to the Author):

Mugat et al. report the biochemical and functional characterisation of the Drosophila piRITS complex. Using a GST pulldown approach they identify Mi-2, MEP-1 and Egg as novel interactors of GST-Gtsf1. These interactions are also probed by co-immunoprecipitation and sucrose gradient fractionation experiments. The authors conclude that piRITS contains the NuRD complex. They observe derepression of transposable elements upon downregulation of Mi-2 or MEP-1 expression. This derepression is

accompanied by a decrease of the the H3K9me3 mark. They then analyse the role of Mi-2 and MEP-1 in piRNA mediated repression using a reporter system and conclude that the NuRD complex is involved in repression and H3K9me3 deposition. Using artificial recruitment of MEP-1 to a reporter gene in germ cells they demonstrate that this is sufficient to decrease reporter gene expression and H3K9 acetylation while simultaneously increasing H3K9me3 levels. Finally, they show colocalisation and co-immunoprecipitation of CHD5, a Mi-2 homolog, and MIWI2 in mouse prospermatogonia suggesting that this interaction is conserved in mammals.

This study addresses timely and important questions: What is the composition of piRITS complexes and how do they contribute to the repression of TEs? The identification of Mi-2 and MEP-1 as piRITS associated factors and their role in maintaining TE repression is novel and of great interest to many in the field. The manuscript is well written and the data generally of high quality.

However, central conclusions about the association of the NuRD complex (as opposed to other Mi-2/MEP-1 assemblies) with piRITS and the involvement of NuRD in piRITS-mediated repression are not yet fully supported by the data. In the following I am suggesting a number of control experiments that could strengthen the manuscript. If the authors can substantiate their claims I would find this manuscript to be of great interest for the readership of Nature Communications.

Major issues:

1. Figure 1: There is conflicting data about the association of MEP-1 with the NuRD complex in the literature. One study reports that MEP-1 is a bona fide subunit of NuRD, another study shows that MEP-1 binds Mi-2 to form the Mec complex but does not associate with NuRD subunits. For this reason it is of particular importance to use careful control experiments and to carefully reconsider the conclusions drawn from the co-immunoprecipitation experiments shown in Figure 1a.

Fig 1a shows that MEP-1 co-immunoprecipitates with Mi-2, CAF1 and Rpd3 supporting the hypothesis that MEP-1 can interact with NuRD in OSC cells. Gtsf1 likewise co-immunoprecipitates with Mi-2, CAF1 and Rpd3 and the authors conclude that Gtsf1 interacts with NuRD. However, CAF1 and Rpd3 also exist in other complexes such as the Sin3 HDAC complex (CAF1 and Rpd3) and PRC2 (CAF1). Thus, an alternative explanation is that Gtsf1 does not interact with NuRD but instead with Mec and other complexes. The experiment lacks a negative control, i.e. all proteins analysed do coprecipitate, raising concerns about the specificity of this IP.

Therefore, the authors should test for co-immunoprecipitation of Sin3 (Sin3 HDAC complex) and E(z) or Su(z)12 (PRC2 complex) as well as for co-IP of NuRD-specific subunits such as MTA-like, MBD-like or p66. This would provide negative controls (Sin3, E(z) and Su(z)12) and also strengthen the conclusion that NuRD (and not only Mec) is part of piRITS.

I do realise that the re-IP experiment presented in Figure 1c supports the author's conclusion that a MEP-1 containing NuRD complex does indeed associate with piRITS. However, the Western blot signals

are very weak and the same specificity concerns (no negative control) apply.

Another general point: In all Figures showing Western Blots / IPs it is of fundamental importance to state how much input (in %) was loaded for each IP. Otherwise it is impossible to judge the relative efficiencies of co-immunoprecipitations which might greatly influence the conclusions drawn. For example, assuming that the same % input was loaded for all IPs shown in Figure 1a, it appears that only a small proportion of Mi-2 is associated with MEP-1 but much larger proportions of CAF1 and Rpd3 (compare intensities of input and IP signals) - this is surprising given that, unlike Mi-2, CAF1 and Rpd3 are components of several additional complexes, does this mean that MEP-1 binds these complexes as well (see above)?

2. The sucrose gradient separation presented in Figure 1b is of limited value. Many of the proteins analysed elute in broad, partially overlapping peaks. This does not really help to delineate the composition of piRITS. In fact, in some ways this data weakens the author's hypothesis, e.g. if NuRD is associated with Gtsf1 then why is Mi-2 not detectable in the Gtsf1 peak fraction (#3). Likewise, why is Gtsf1 not detectable in the MEP-1 peak fraction (#5)? Moreover, Mi-2 and MEP-1 do not peak in the same fraction arguing that MEP-1 is not a bona fide NuRD subunit. In order to demonstrate that co-fractionation is caused by interaction the authors should perform co-immunoprecipitation experiments from sucrose gradient fractions.

3. Figure 3 compares the apiRNA-mediated repression of a reporter gene in control cells and in cells where MEP-1 or Mi-2 have been depleted by RNAi. In control cells apiRNA expression results in a 40% repression (Figure 3b) that is reduced to a 20% repression when Mi-2 is depleted. I am assuming this means that under Mi-2 KD conditions the remaining reporter gene expression is 80%. In Figure 3c, reporter gene expression is measured in the absence of apiRNA and here Mi-2 depletion reduces reporter gene expression to a similar extent (to about 80%). Why do the authors think that reporter gene repression by apiRNA and by Mi-2 depletion are not simply additive effects? I find this very confusing. I suggest that in both figures „RNA levels“ should be plotted rather than „% repression“ in Fig. 3 a and „RNA levels“ in Fig. 3 b to allow for a better comparison.

The authors conclude that the effects shown in Figure 3 are mediated by NuRD. However, as explained above they could be due to the Mec complex. The authors should either include an RNAi KD of a NuRD-specific subunit or rephrase their conclusions.

4. In Figure 4 that authors use a system that allows artificial tethering of MEP1 to a reporter gene in the germline. They find that this reduces reporter gene expression, decreases H3K9ac and increases H3K9me3. They conclude that these results support the hypothesis that MEP1/NuRD mediates repression and chromatin changes as part of the piRITS complex. This conclusion is not supported by the data. Many repressors tethered to the reporter gene would result in reduced expression and, accordingly, a decrease in activating and an increase in repressive histone modifications. A good experiment to link the observed effects to the piRITS complex would be to perform the analysis in an

Egg deficient background (Egg being the piRITS subunit responsible for H3K9 methylation by the complex). If this rescues reporter gene expression and histone modification changes then this would constitute strong support for the author's model. Alternatively, the authors could performed CHIP to demonstrate that piRITS components (e.g. Egg) are recruited by MEP-1.

5. Figure 5 shows colocalisation of CHD5 and MIWI2 in spermatogonia and co-immunoprecipitation of both proteins from extracts. Based on this the authors propose that the interaction of NuRD and piRITS is conserved in mammals. I find this data to be too preliminary to draw such a conclusion. First, as in Figure 1, there is no control to show that the CHD5 antibody does not unspecifically precipitate nuclear proteins. Second, to strengthen the hypothesis I suggest to expand the co-localisation and co-immunoprecipitation studies to other subunits of NuRD and piRITS.

6. The authors propose that NuRD/piRITS act together to repress genes in a piRNA dependent manner. Is it possible to provide more direct support for this notion by chromatin IP of NuRD subunits, Egg etc (to show that they colocalise at the same chromatin sites)?

Minor issue:

6. Figure 1 b: Egg runs as a double band (input), only the faster migrating polypeptide appears to efficiently associate with MEP1. This is different from the results displayed in Figures 1 a and 1 c - the authors should discuss these discrepancies.

Reviewer #3 (Remarks to the Author):

Mugat et al. describe a series of experiments that implement the NuRD complex in the Piwi-induced H3K9 methylation of piRNA targets. This is an interesting, and important research topic, as it still remains very much unclear how Piwi-piRNA complexes trigger histone modification. Two factors have previously been implicated: Gtsf1 and Panx. In addition the histone methyl transferase Eggless (Egg) has been implicated. The authors propose that the NuRD complex bridges the gap between the Piwi-Gtsf1-Panx complex and Egg. If this is conclusively shown, this certainly warrants a paper in Nature Comm. However, with the results that are currently shown, I am not certain this has been proven. I believe a number of additional controls are required to make this statement.

Specific issues:

1) The GST-pull-down of Gtsf1 is very noisy. A tremendous amount of proteins are pulled down, making it difficult to assess the specificity. Focussing on NuRD is really cherry-picking, building on the fact that indeed NuRD components have already been implicated in the silencing through genetic experiments (mentioned by the authors). The authors now describe an endogenously tagged Gtsf1 allele in OSC cells, so it would be great if that can be used to maybe clean-up the IP mass spec.

2) The glycerol gradients are not very convincing in showing a complex containing Piwi, Gtsf1, Panx, NuRD and Egg. Many of the peaks are so broad that an overlap is given by default. This may be inherent

to the fact that NuRD is much more broadly involved in chromatin modification, but the fact remains that the results are very hard to interpret. Is it not possible to take fractions where the various factors co-elute, and see if they continue to do so on a different column?

3) The coIPs are always controlled by IgG. I would like to see more stringent controls, such as IP with the HA antibody in a background lacking the HA-fusion protein (or ideally, another HA-tagged fusion protein). A crucial experiment would also be to demonstrate that the Piwi-NuRD interaction is lost upon loss of Gtsf1 or Panx.

4) The authors should test if the tethering effect of Panx also requires NuRD.

5) In the discussion, the authors mention a potential role for NuRD in the feed-forward amplification of piRNAs. At the same time they show NuRD does not affect piRNA levels. These two statements are in stark contrast. Do the experiments really show that NuRD is not needed for piRNA generation? Piwi-piRNA complexes can be very stable, and I am not convinced the experiments take this into account. In other words, are the knock-downs performed long enough? As control, the authors should therefore also knock down, for instance, Zucchini, or another established piRNA biogenesis factor.

We would like to sincerely thank the reviewers for their insightful comments and constructive suggestions. We hope that they will be convinced by our extensively revised manuscript showing now that Piwi transiently interacts with 3 novel co-repressors, Rpd3, Mi-2 and MEP-1, for the piRNA-dependent heterochromatin formation that enforces TE repression.

Reviewer #1

Here the authors present the results showing the roles of NuRD subunits in TE repression. It is pretended that the specific piRITS nuclear complex has been isolated using IP with Gtsf-FLAG-HA expression. Mass spec analysis of GST-GTSF IP shows significant enrichment of several putative components of this complex, including two NuRD subunits as well as a lot of some other associated proteins (Fig S1b). Re-immunoprecipitation of this IP using Ab against MEP-1 confirms the presence of putative complex components, but re-mass spec analysis of re-IP is needed to show further complex purification as well as some hints to reveal the stoichiometric presence of putative RITS components. Possibly CL approach and MS analysis would be desirable to analyze re-precipitate.

In light of the referees' comments, we deeply reinvestigated the identification of piRITS corepressors and a series of biochemical experiments presented in revised Fig. 1 now provide further evidence for sub-stoichiometric interactions between Gtsf1 and the 3 NuRD sub-units (Rpd3 (HDAC), Mi-2, and MEP-1):

- Pull-down experiments using recombinant GST-Gtsf1 followed by mass spectrometry analysis led to the identification of the Rpd3 histone deacetylase (HDAC1 homolog), the Mi-2 nucleosome remodeling ATPase and the MEP-1 Krüppel-type zinc-finger protein as putative Gtsf1 partners (revised Supplementary. Fig. 1).

- Overexpressed GFP-tagged Gtsf1 construct in OSC cells co-immunoprecipitates with these three interactors (revised Fig. 1a).

- To study the binding efficiency of Gtsf1 with Rpd3, Mi-2, or MEP-1, we performed a semi-quantitative luminescence-based co-immunoprecipitation between overexpressed Gtsf1 and tagged prey (Rpd3, Mi-2, or MEP-1) in S2 cells (Revised Fig. 1b,c). This approach revealed that Gtsf1 pulls down Rpd3, Mi-2, or MEP-1 significantly more than the negative control.

- To prevent artificial interactions due to overexpression approaches, we now present co-IPs of endogenous proteins using specific antibodies against Piwi and MEP-1 in OSC cells (Revised Fig. 1d) that also confirm the interactions between Piwi/Gtsf1 and Mi-2, MEP-1, Rpd3 and its associated p55 protein. In contrast, no interaction could be detected with Pc, a PRC1 Polycomb complex component.

- As explained below to reviewer#3 (point 3), we had to withdraw the re-IP data because they relied on a non-specific anti-HA antibody.

The author's attempts in favor to detect the existence of this complex using glycerol centrifugation were unsuccessful (Fig.1c). So today it is possible to discuss only the transient co-association of the studied components.

The sentence (lines 93-95) related to the complex structure discussion looks inappropriate.

As mentioned by the referee, the glycerol gradient assay was not very successful and the data were therefore removed from the revised version. We now show a size fractionation experiment using a Superpose 6 liquid chromatography (Revised Fig. 2a). This approach produced improved resolution profiles of multi-protein modules from OSC nuclear extracts. MEP-1 and Mi-2 peak in the same fraction in OSC cells as previously reported in Kc167 cells (Kunert et al., 2009). The fractionation profile indicates the existence of at least 2 distinct stable complexes: dMec (Mep-1 and Mi-2) and a Piwi/Gtsf1 subcomplex. In contrast, Rpd3/p15 that is known to belong to several repressor complexes elutes in broad peaks.

This experiment clearly indicates that Piwi/Gtsf1 do not form a single stable complex with MEP-1, Mi-2 and Rpd3. Instead a stable Piwi/Gtsf1 subunit was found to co-elute away from stable preformed complexes Mi-2/MEP-1, and putative NuRD and Rpd3/p15 complexes. In addition, neither Piwi nor MEP-1 were able to pull down the dNuRD subunit MTA1-like suggesting that the *bona fide* dNuRD complex is not associated to Piwi (Revised Fig. 2b). Thanks to the reviewer comments, we are now considering a transient co-association between at least 3 stable sub-units involved in the Piwi-associated silencing machinery that we no longer call "a silencing complex". The text was modified accordingly.

Fig.4 demonstrates only an ability of HDAC recruitment by MEP-1 as an itself peculiar NURD functional property. Fig.6S presents Egg/SetDB as a component of pi-RITS complex in the absence of a required direct experiment demonstrating this interaction. Position of Egg peptide is not even represented in Fig.S2b. Thus the author's statement in the abstract that they «present the first biochemical and genetic characterization of the Drosophila piRITS complex» is exaggerated.

The distinct part of this paper concerns the role of NuRD subunits in TE repression, but the relation of these results to the existence of postulated pi-RITS complex remains under question. The observation that piRNA induced krimp repression is similarly dependent upon Piwi, MEP-1 and Mi-2 KDs is an indication but not the evidence of postulated complex activity. The same comment concerns the observation of MEP-1 recruitment of HDAC. Thus, these observations themselves are of some interest, but they cannot serve as cardinal proof of the existence of a complex.

To further demonstrate the role of these co-repressors in TE repression, the revised version now provides additional genetic approaches showing that Rpd3, Mi-2 and MEP-1 are all required for piRNA-mediated TE transcriptional repression.

- RNA-seq experiments were performed in collaboration with the Siomi lab after Rpd3, Mi-2 or Piwi knockdown in OSC cells. Revised Fig. 3a, 3b and Supplementary Fig. 3c reveal that the profile of derepressed TE families after Rpd3 RNAi is the same as after Mi-2 or Piwi RNAi. These data were confirmed by qRT-PCR (Revised Fig. 3c). In contrast, supporting our biochemical data, the dNuRD complex *per se* is not involved in TE repression since depletion of MTA1-like or MBD did not reveal any mRNA increase for endogenous TEs (Revised Supplemental Fig. 3d). We also show that the derepression in OSCs correlates with a decrease of the H3K9me3 repressive mark and an increase of the H3K9acetyl active mark on endogenous TEs (Revised Fig. 3d,e).

- Depletion of Mi-2 and MEP-1 *in vivo* in somatic tissues also led to a significant TE derepression (Revised Fig. 4).

Finally, it has been recently reported that the SUMO E3 ligase, Su(var)2-10, is involved in the piRNA transcriptional repression via its binding with SetDB1 (Ninova et al., 2019). In the revised version, we now show that MEP-1 physically interacts with Su(var)2-10 and SetDB1 via the C-terminal domain of MEP-1 (Revised Figures 6a and 6b). These new biochemical approaches help us to propose a model in which MEP-1/Mi-2 and Su(var)2-10 interact together and act as co-repressors of TE transcriptional repression.

Minor comments

Fig 1a Absence of designations above the strips

We have changed the figures accordingly.

It is unclear why fraction 4 content (Fig.1c) is selected to evaluate whether any component will be related to a putative complex.

This figure has been replaced by Fig. 2a in the revised version and we are not talking about a whole silencing complex anymore.

Fig.2a. The results related to HetA are not clearly presented, picture replacement required

These data have been removed from the revised version of the manuscript and the revised Fig. 4 illustrates the function of MEP-1, Mi-2 and Piwi on TE repression *in vivo*, in ovarian somatic tissues.

Reviewer #2

Mugat et al. report the biochemical and functional characterisation of the Drosophila piRITS complex. Using a GST pulldown approach they identify Mi-2, MEP-1 and Egg as novel interactors of GST-Gtsf1. These interactions are also probed by co-immunoprecipitation and sucrose gradient fractionation experiments. The authors conclude that piRITS contains the

IGH, 141 rue de la Cardonille, 34396 Montpellier Cedex 5 France
Phone: +33 (0)434 359 944 – Fax: +33 (0)434 359 949 – E-mail: severine.chambeyron@igh.cnrs.fr
<http://www.igh.cnrs.fr>

NuRD complex. They observe derepression of transposable elements upon downregulation of Mi-2 or MEP-1 expression. This derepression is accompanied by a decrease of the H3K9me3 mark. They then analyse the role of Mi-2 and MEP-1 in piRNA mediated repression using a reporter system and conclude that the NuRD complex is involved in repression and H3K9me3 deposition. Using artificial recruitment of MEP-1 to a reporter gene in germ cells they demonstrate that this is sufficient to decrease reporter gene expression and H3K9 acetylation while simultaneously increasing H3K9me3 levels. Finally, they show colocalisation and co-immunoprecipitation of CHD5, a Mi-2 homolog, and MIWI2 in mouse prospermatogonia suggesting that this interaction is conserved in mammals.

This study addresses timely and important questions: What is the composition of piRITS complexes and how do they contribute to the repression of TEs? The identification of Mi-2 and MEP-1 as piRITS associated factors and their role in maintaining TE repression is novel and of great interest to many in the field. The manuscript is well written and the data generally of high quality.

However, central conclusions about the association of the NuRD complex (as opposed to other Mi-2/MEP-1 assemblies) with piRITS and the involvement of NuRD in piRITS-mediated repression are not yet fully supported by the data. In the following I am suggesting a number of control experiments that could strengthen the manuscript. If the authors can substantiate their claims I would find this manuscript to be of great interest for the readership of Nature Communications.

Major issues:

1. Figure 1: There is conflicting data about the association of MEP-1 with the NuRD complex in the literature. One study reports that MEP-1 is a bona fide subunit of NuRD, another study shows that MEP-1 binds Mi-2 to form the Mec complex but does not associate with NuRD subunits. For this reason it is of particular importance to use careful control experiments and to carefully reconsider the conclusions drawn from the co-immunoprecipitation experiments shown in Figure 1a.

Fig 1a shows that MEP-1 co-immunoprecipitates with Mi-2, CAF1 and Rpd3 supporting the hypothesis that MEP-1 can interact with NuRD in OSC cells. Gtsf1 likewise co-immunoprecipitates with Mi-2, CAF1 and Rpd3 and the authors conclude that Gtsf1 interacts with NuRD. However, CAF1 and Rpd3 also exist in other complexes such as the Sin3 HDAC complex (CAF1 and Rpd3) and PRC2 (CAF1). Thus, an alternative explanation is that Gtsf1 does not interact with NuRD but instead with Mec and other complexes. The experiment lacks a negative control, i.e. all proteins analysed do coprecipitate, raising concerns about the specificity of this IP.

Therefore, the authors should test for co-immunoprecipitation of Sin3 (Sin3 HDAC complex) and E(z) or Su(z)12 (PRC2 complex) as well as for co-IP of NuRD-specific subunits such as

MTA-like, MBD-like or p66. This would provide negative controls (Sin3, E(z) and Su(z)12) and also strengthen the conclusion that NuRD (and not only Mec) is part of piRITS.

I do realise that the re-IP experiment presented in Figure 1c supports the author's conclusion that a MEP-1 containing NuRD complex does indeed associate with piRITS. However, the Western blot signals are very weak and the same specificity concerns (no negative control) apply.

Another general point: In all Figures showing Western Blots / IPs it is of fundamental importance to state how much input (in %) was loaded for each IP. Otherwise it is impossible to judge the relative efficiencies of co-immunoprecipitations which might greatly influence the conclusions drawn. For example, assuming that the same % input was loaded for all IPs shown in Figure 1a, it appears that only a small proportion of Mi-2 is associated with MEP-1 but much larger proportions of CAF1 and Rpd3 (compare intensities of input and IP signals) - this is surprising given that, unlike Mi-2, CAF1 and Rpd3 are components of several additional complexes, does this mean that MEP-1 binds these complexes as well (see above)?

As pointed out by the reviewer there are conflicting data about the association of MEP-1 with the NuRD complex in the literature. Based on the literature (Reddy et al., 2010; Kunert et al., 2009), we initially thought that Rpd3-associated MEP-1 was part of the canonical NuRD complex. In light of our revised data, we now show that in OSC cells, MEP-1 can interact with Rpd3 in a NuRD-independent manner. In other words, we now favour the alternative proposed by this reviewer (*Gtsf1 does not interact with dNuRD but instead with dMec and other complexes*).

First, we deeply reinvestigated the identification of piRITS corepressors biochemically and revised Figure 1 now shows the interaction between Gtsf1 and the 3 NuRD sub-units (Rpd3 (HDAC), Mi-2, and MEP-1): see above our first answer to reviewer#1

As suggested by reviewer#2, we added negative controls for the IPs (see Revised Fig. 1 and Fig. 2). Moreover, the IPs have been performed with controlled input amounts and the % input is indicated in Fig. 1 and Fig. 2.

Second, we now show that neither Piwi nor MEP-1 were able to pull down the dNuRD subunit MTA1-like, suggesting that the *bona fide* dNuRD complex is not associated to Piwi (Revised Fig. 2b). In addition, knocking down MTA1-like or MBD-like, in OSC cells did not reveal any mRNA increase for endogenous TEs nor for the *ex* gene (Supplementary Fig. 3d) whereas most of the TEs derepressed upon Piwi depletion were also derepressed upon depletion of Mi-2 or Rpd3, thus confirming genetically, that the dNuRD complex *per se* is not involved in TE repression.

2. The sucrose gradient separation presented in Figure 1b is of limited value. Many of the proteins analysed elute in broad, partially overlapping peaks. This does not really help to delineate the composition of piRITS. In fact, in some ways this data weakens the author's hypothesis, e.g. if NuRD is associated with Gtsf1 then why is Mi-2 not detectable in the Gtsf1 peak fraction (#3). Likewise, why is Gtsf1 not detectable in the MEP-1 peak fraction (#5)? Moreover, Mi-2 and MEP-1 do not peak in the same fraction arguing that MEP-1 is not a bona fide NuRD subunit. In order to demonstrate that co-fractionation is caused by interaction the authors should perform co-immunoprecipitation experiments from sucrose gradient fractions.

See our second answer to reviewer#1.

3. Figure 3 compares the apiRNA-mediated repression of a reporter gene in control cells and in cells where MEP-1 or Mi-2 have been depleted by RNAi. In control cells apiRNA expression results in a 40% repression (Figure 3b) that is reduced to a 20% repression when Mi-2 is depleted. I am assuming this means that under Mi-2 KD conditions the remaining reporter gene expression is 80%. In Figure 3c, reporter gene expression is measured in the absence of apiRNA and here Mi-2 depletion reduces reporter gene expression to a similar extent (to about 80%). Why do the authors think that reporter gene repression by apiRNA and by Mi-2 depletion are not simply additive effects? I find this very confusing. I suggest that in both figures „RNA levels“ should be plotted rather than „% repression“ in Fig. 3 a and „RNA levels“ in Fig. 3 b to allow for a better comparison.

We understand that presenting the effect of apiRNAs by a percentage of repression may be somewhat confusing. We are now presenting the data in a more usual and intuitive way as suggested by the reviewer. Instead of % of repression, the revised Fig. 5b displays the fold change induced by apiRNA treatment (i.e. RNA level after apiRNA treatment normalized by that of the corresponding GFP treatment control). Moreover, in the revised Fig. 5c, RNA levels of all independent experiments are plotted directly. This figure shows that none of the tested proteins is involved in *krimp* repression in the absence of apiRNAs : no derepression is observed but only a small decrease in *krimp* expression is caused by some of the tested siRNAs. No additive effect of the siRNA and the apiRNA is observed since, in this figure, we do not observe a decrease in *krimp* RNA level between “apiRNA+siMi-2 “and “apiRNA+sictrl“.

The authors conclude that the effects shown in Figure 3 are mediated by NuRD. However, as explained above they could be due to the Mec complex. The authors should either include an RNAi KD of a NuRD-specific subunit or rephrase their conclusions.

As previously mentioned, we now show that neither Piwi nor MEP-1 were able to pull down the NuRD subunit MTA1-like suggesting that the bona fide NuRD complex is not associated to Piwi (Revised Fig. 2b). In addition, knocking down MTA1-like or MBD, in OSC cells did not reveal any mRNA increase for endogenous TEs nor for the *ex* gene (Supplementary Fig.

3d), thus confirming genetically, that the dNuRD complex *per se* is not involved in TE repression.

According to the reviewer's comments, we now revised the conclusion: "Taken together, these data suggest that Mi-2 and MEP-1 interact with Piwi not as components of a dNuRD but as a dMec preformed module (p7).

4. In Figure 4 that authors use a system that allows artificial tethering of MEP1 to a reporter gene in the germline. They find that this reduces reporter gene expression, decreases H3K9ac and increases H3K9me3. They conclude that these results support the hypothesis that MEP1/NuRD mediates repression and chromatin changes as part of the piRITS complex. This conclusion is not supported by the data. Many repressors tethered to the reporter gene would result in reduced expression and, accordingly, a decrease in activating and an increase in repressive histone modifications. A good experiment to link the observed effects to the piRITS complex would be to perform the analysis in an Egg deficient background (Egg being the piRITS subunit responsible for H3K9 methylation by the complex). If this rescues reporter gene expression and histone modification changes then this would constitute strong support for the author's model. Alternatively, the authors could performed ChIP to demonstrate that piRITS components (e.g. Egg) are recruited by MEP-1.

We tried many times to perform ChIP experiments using different antibodies and several protocols but we failed to chromatin-immunoprecipitate Egg and the co-repressors described in this manuscript. Moreover, in the revised version, we have added new data showing that MEP-1 and Mi-2 interact with Su(var)2-10 (Revised Fig. 6). This protein has been described as an Egg interactor (Ninova et al., 2019). These new data indicate why MEP-1 and Mi-2 interact with Egg.

5. Figure 5 shows colocalisation of CHD5 and MIWI2 in spermatogonia and co-immunoprecipitation of both proteins from extracts. Based on this the authors propose that the interaction of NuRD and piRITS is conserved in mammals. I find this data to be too preliminary to draw such a conclusion. First, as in Figure 1, there is no control to show that the CHD5 antibody does not unspecifically precipitate nuclear proteins. Second, to strengthen the hypothesis I suggest to expand the co-localisation and co-immunoprecipitation studies to other subunits of NuRD and piRITS.

As highlighted by the reviewer, these data were to preliminary and were thus removed from the revised version. However, we just would like to mention to the reviewer that we have performed the reciprocal IP using CHD5 antibody and we were able to detect Miwi-2 pulled down by CHD5 (see Figure below).

6. The authors propose that NuRD/piRITS act together to repress genes in a piRNA dependent manner. Is it possible to provide more direct support for this notion by chromatin IP of NuRD subunits, Egg etc (to show that they colocalise at the same chromatin sites)?

As previously mentioned, we tried many times to perform ChIP experiments using different antibodies and several protocols but we failed to immunoprecipitate the chromatin associated to dMec or Egg.

Minor issue:

6. Figure 1 b: Egg runs as a double band (input), only the faster migrating polypeptide appears to efficiently associate with MEPI. This is different from the results displayed in Figures 1 a and 1 c - the authors should discuss these discrepancies.

The biochemical approaches have been changed and we do not see a discrepancy anymore. Moreover, the post-translational modification of nuclear Egg has been recently reported (Osumi et al., 2019). These two forms, present in the input, correspond to nuclear Egg being either ubiquitinated and phosphorylated or ubiquitinated only.

Reviewer #3 (Remarks to the Author):

Mugat et al. describe a series of experiments that implement the NuRD complex in the Piwi-induced H3K9 methylation of piRNA targets. This is an interesting, and important research topic, as it still remains very much unclear how Piwi-piRNA complexes trigger histone modification. Two factors have previously been implicated: Gtsf1 and Panx. In addition the histone methyl transferase Eggless (Egg) has been implicated. The authors propose that the NuRD complex bridges the gap between the Piwi-Gtsf1-Panx complex and Egg. If this is conclusively shown, this certainly warrants a paper in Nature Comm. However, with the results that are currently shown, I am not certain this has been proven. I believe a number of additional controls are required to make this statement.

We would like to warmly thank this reviewer for his very insightful comments, especially the suggested controls that have significantly improved the quality of our manuscript. As explained above in the answer to reviewer#2, our new data suggest that the new Piwi-associated silencing factors that are disclosed by our work (Rpd3, Mi-2 and Mep-1), are not components of a preformed dNuRD complex but rather seem to belong to two independent subcomplexes. One of them, the dMec, contains the chromatin remodeler Mi-2 and a protein of unknown function,

MEP-1. Since the submission of the first version, a new factor, Su(var)2-10, has been reported to interact with Piwi, Gtsf1 and Egg (Ninova et al., 2019). We show now that dMec does interact with Su(var)2-10 and that the C-terminus of MEP-1 is directly implicated in this interaction. This new data reveals that the Piwi-dependent TE transcriptional repression hijacks the SUMO-dependent transcriptional silencing machinery, including MEP-1, Mi-2 and Su(var)2-10, that is usual guided by DNA-bound SUMO-modified transcription factors (Stielow et al., 2007).

Specific issues:

1) The GST-pull-down of Gtsf1 is very noisy. A tremendous amount of proteins are pulled down, making it difficult to assess the specificity. Focussing on NuRD is really cherry-picking, building on the fact that indeed NuRD components have already been implicated in the silencing through genetic experiments (mentioned by the authors). The authors now describe an endogenously tagged Gtsf1 allele in OSC cells, so it would be great if that can be used to maybe clean-up the IP mass spec.

In light of the comments of point 3 of this reviewer on the fact that “the coIPs are always controlled by IgG”, we performed IP with the HA antibody in a background lacking the HA-fusion protein. We have observed an unspecific Piwi binding on anti-HA beads even when using non-HA tagged nuclear extracts. That is the reason why we could not perform the experiment proposed here to clean-up the IP mass spec.

All the part of the manuscript using this HA antibody (Revised Fig. 1) has been removed and new biochemical approaches with controls confirming the previous data have been added.

All the new biochemical approaches performed in the revised version have been listed in the answer to the reviewer#1's comments.

2) The glycerol gradients are not very convincing in showing a complex containing Piwi, Gtsf1, Panx, NuRD and Egg. Many of the peaks are so broad that an overlap is given by default. This may be inherent to the fact that NuRD is much more broadly involved in chromatin modification, but the fact remains that the results are very hard to interpret. Is it not possible to take fractions where the various factors co-elute, and see if they continue to do so on a different column?

As mentioned by this referee, the glycerol gradient assay was not very resolutive and it was removed from the revised version. We now show a size fractionation experiment using a Superpose 6 column (Revised Fig. 2a). This approach produced improved resolution profiles of multi-protein modules from OSC nuclear extracts. MEP-1 and Mi-2 peak in the same fraction in OSC cells as previously reported in dMec-containing Kc167 cells (Kunert et al., 2009). The fractionation profile indicates the existence of at least 2 stable complexes: dMec (Mep-1 and Mi-2) and a Piwi/Gtsf1 subcomplex. In contrast, Rpd3/p15 that is known to belong to several repressor complexes elutes in broad peaks.

This experiment clearly indicates that Piwi/Gtsf1 do not form a single stable complex with MEP-1, Mi-2 and Rpd3. Instead a stable Piwi/Gtsf1 subunit was found to co-elute away from three stable preformed complexes the Mi-2/MEP-1-containing dMec, the putative NuRD and various Rpd3/p15-containing complexes. In addition, neither Piwi nor MEP-1 were able to pull down the dNuRD subunit MTA1-like, suggesting that the *bona fide* dNuRD complex is not associated to Piwi (Revised Figure 2b). Thanks to the reviewer comments, we are now considering a transient co-association of at least 3 stable sub-units involved in the Piwi-associated silencing machinery that we no longer call “a silencing complex”. The text was modified accordingly.

3) *The coIPs are always controlled by IgG. I would like to see more stringent controls, such as IP with the HA antibody in a background lacking the HA-fusion protein (or ideally, another HA-tagged fusion protein). A crucial experiment would also be to demonstrate that the Piwi-NuRD interaction is lost upon loss of Gtsf1 or Panx.*

As previously mentioned, the control IP in a background lacking the HA-fusion protein has been done and revealed that Piwi was pulled down by the HA antibody even in this negative background (Revised Supplementary Fig. 2c). This experiment has been removed and negative controls like Pc and MTA-1 have been added.

Moreover, GFP nanobodies have been used to pull down the overexpressed Gtsf1-GFP protein and the stringency of such an experiment was checked by GFP-expressing nuclear extracts used as negative controls.

Whether Gtsf1 or Panx are required for the Piwi-dMec interaction is still unknown. In other words, this work concentrates on the composition of the Piwi silencing machinery, but not on the way it interacts with the piRNA-guided target recognition complex.

4) The authors should test if the tethering effect of Panx also requires NuRD.

We chose the lacI-lacO system to artificially guide corepressors to a reporter. This system allowed us to show that MEP-1, a protein of unknown function, is able to recruit some H3K9 deacetylase and trimethylase activities resulting in the reporter repression. As a positive control, we used Panx tethered to the DNA, too. As suggested by the reviewer, we checked if this Panx-mediated repression was MEP-1-dependent or not. After the MEP-1 RNAi, Panx repression was still observed. However, it was not impaired by HP-1 depletion either, our positive control. By contrast the silencing is known to be HP-1-dependent when Panx is tethered to the nascent RNA (Zhao et al., 2019). We therefore cannot answer the reviewer conclusively although the new data on Suvar2-10 and dMec predict a positive answer.

5) In the discussion, the authors mention a potential role for NuRD in the feed-forward amplification of piRNAs. At the same time they show NuRD does not affect piRNA levels. These two statements are in stark contrast. Do the experiments really show that NuRD is not needed for piRNA generation? Piwi-piRNA complexes can be very stable, and I am not

convinced the experiments take this into account. In other words, are the knock-downs performed long enough? As control, the authors should therefore also knock down, for instance, Zucchini, or another established piRNA biogenesis factor.

This confusing paragraph has been withdrawn from the discussion. Indeed, to test the effect of Rpd3, Mi-2 and Mep-1 on piRNA biogenesis, we depleted these proteins in somatic ovarian cells. By contrast, a potential role for NuRD in the feed-forward amplification of piRNAs is only expected to occur in the germ cells where the RDC complex is expressed.

REVIEWERS' COMMENTS:

Reviewer #1 (Remarks to the Author):

There are two different sections to this article. In the first of these, the physical interactions of the Rpd3, MEP-I, and Mi-2 proteins with Piwi are considered. CO-IP experiments indicated the possibility of such interactions, confirmed by reciprocal CO-IP. The authors rightly emphasize that the Rpd3 / HDAC function was not previously identified as a component of the piRNA-Piwi dependent silencing complex. At the same time, the authors failed to demonstrate association of these components with Piwi using gel filtration and no attempts to perform cross-link-IP have been performed to reveal this association possibility. The presence of preformed Mi-2-MEP-1 complex is demonstrated only indirectly, the evidence of its existence as a stable subunit (line 25 in abstract) remains elusive. The demonstration of MEP-1 association with Su(var)2-10 looks interesting indicating the participation of remodeling complex in TE silencing as well as a continuation of the sumoylation studies in this type of silencing following the recent work of Ninova et al 2019. In general, results, indicating the participation of only certain components of the classic NuRD complex in the TE silencing, are of undoubted interest.

The second part of this paper concerns the studies of KD impacts on TE silencing and for some reason these results are not reflected in the abstract.

My minor comments. In the abstract, the authors should note exactly and clearly the main results that they revealed. Possible confusion with data recently acquired by another group (Ninova et al 2019) should be avoided, especially by the last abstract sentence. The similar comment applies to the introduction, in which the authors need to determine by evaluating the results of Ninova et al as suggested or proven/shown/revealed. The results of gel filtration experiments do not deserve such a lengthy presentation.

Reviewer #2 (Remarks to the Author):

Major points:

I have had concerns about the specificity of the reported biochemical interactions between Gtsf1, Piwi, dMi-2 and various dMi-2-associated proteins and the author's interpretation that Gtsf1 and Piwi biochemically and functionally interact with the dNuRD complex. The revised manuscript contains a greatly expanded interactome and interaction analysis and includes essential negative controls. The new data substantiates the reported interactions but has also changed the authors' conclusions. They no longer propose that Gtsf1/Piwi complex with dNuRD but instead that they complex with an dMi-2/dMEP1 complex and, independently, with dRPD3/p55. This conclusion is now much better supported by the data shown.

My second major concern was the representation and quantification of the experiment shown in Figure

5 (effect of depletion of various proteins on the apiRNA-mediated repression of the krimp gene) which made it difficult for me to interpret the data. In the revised version of the manuscript, the representation of the data is much improved and, as a consequence, the data is much easier to interpret. However, I fail to see a significant effect of dMEP1, dMi-2, dRpd3 and Piwi depletion on apiRNA-mediated repression. The krimp RNA levels after apiRNA expression (Figure 5c, "+ apiRNA" bars) range from approx. 1.0 to 1.4. The fold changes (Figure 5b; krimp RNA levels in presence of apiRNA over control (GFP)) appear more substantial (increasing from approx. 0.4 in the control to a maximum of 1.0 in the siRpd3 experiment) but these changes are largely driven by the apiRNA-independent repression of krimp illustrated in Figure 5c ("- apiRNA" bars). The general problem with this assay is that apiRNA-mediated repression of krimp in the control is already weak (from 1.8 down to 1.1, i.e. less than 2-fold). Given the small effect sizes I find it difficult to disentangle apiRNA-dependent from apiRNA-independent effects (all knockdowns reduce krimp expression even in the absence of apiRNA). The authors write that (line 239) "These data show that these 3 proteins are as important as Piwi for efficient apiRNA-mediated repression in this system." but even Piwi knockdown appears to increase krimp RNA levels in the presence of apiRNA only from 1.1 to 1.3 (Figure 5c)... I do admit that I am not an expert on this type of assay, so my concerns about the significance of the effects might be unfounded. At the very least, the authors should discuss the effect sizes in the text and explain why they represent indeed "significant" (line 238) changes.

Minor point:

New dual-luciferase co-immunoprecipitation assays are included. These rely on equal expression of the luciferase fusion proteins. I suggest that the authors document the expression levels of the various constructs used, in particular in cases where no interaction was detected.

Reviewer #3 (Remarks to the Author):

The authors did a good job improving the manuscript, and now present a much more balanced story. Unfortunately, this also led to a significant down-tuning of the statements on complexes, as the interactions that are newly uncovered do not appear to be found in stable complexes. Of course, this can always be due to experimental conditions. A nice co-elution would have made the story more straightforward, but it is as it is. The question is whether the evidence implicating these new components in piRNA driven silencing is strong enough, and I would say it is. The discussion also nicely places the findings into perspective. The only finding that is not well discussed is the fact that in vivo Rpd3 depletion did not have much of an effect. Hence, for this protein, the evidence that it really acts in the Piwi pathway is significantly weaker. This is not reflected in the discussion.

Answer to REVIEWERS' COMMENTS:

Reviewer #1 (Remarks to the Author):

There are two different sections to this article. In the first of these, the physical interactions of the Rpd3, MEP-I, and Mi-2 proteins with Piwi are considered. CO-IP experiments indicated the possibility of such interactions, confirmed by reciprocal CO-IP. The authors rightly emphasize that the Rpd3 / HDAC function was not previously identified as a component of the piRNA-Piwi dependent silencing complex. At the same time, the authors failed to demonstrate association of these components with Piwi using gel filtration and no attempts to perform cross-link-IP have been performed to reveal this association possibility. The presence of preformed Mi-2-MEP-1 complex is demonstrated only indirectly, the evidence of its existence as a stable subunit (line 25 in abstract) remains elusive. The demonstration of MEP-1 association with Su(var)2-10 looks interesting indicating the participation of remodeling complex in TE silencing as well as a continuation of the sumoylation studies in this type of silencing following the recent work of Ninova et al 2019. In general, results, indicating the participation of only certain components of the classic NuRD complex in the TE silencing, are of undoubted interest.

The second part of this paper concerns the studies of KD impacts on TE silencing and for some reason these results are not reflected in the abstract.

My minor comments. In the abstract, the authors should note exactly and clearly the main results that they revealed. Possible confusion with data recently acquired by another group (Ninova et al 2019) should be avoided, especially by the last abstract sentence. The similar comment applies to the introduction, in which the authors need to determine by evaluating the results of Ninova et al as suggested or proven/shown/revealed. The results of gel filtration experiments do not deserve such a lengthy presentation.

We agree with the three following comments:

-“the existence of a stable Mi-2-MEP-1 complex remains elusive”.

We therefore deleted the word “stable” accordingly in the abstract.

- "Possible confusion with data recently acquired by another group (Ninova et al 2019) should be avoided, especially by the last abstract sentence "

We added "in addition to the Su(var)2-10 SUMO ligase." at the end of the abstract.

-“ similar comment applies to the introduction, in which the authors need to determine by evaluating the results of Ninova et al as suggested or proven/shown/revealed."

We believe that the revised version of the Introduction now clearly indicates that we have confirmed two of the Su(var)2-10 interactors previously reported by Ninova et al and revealed a new one, MEP-1. *“A physical link between these two modules might be provided by Su(var)2-10 that we found to interact with MEP-1 in both S2 and ovarian somatic cells, in addition to its previously reported Piwi and SetDB1 partners²⁴.”*

By contrast, we disagree with the comment " The second part of this paper concerns the studies of KD impacts on TE silencing and for some reason these results are not reflected in the abstract." Indeed, we feel that the second part of the paper was actually reflected in the following sentence " Together with the histone deacetylase Rpd3, this module is involved in the piRNA-dependent TE silencing, correlated with H3K9 deacetylation and trimethylation."

Reviewer #2 (Remarks to the Author):

Major points:

I have had concerns about the specificity of the reported biochemical interactions between Gtsf1, Piwi, dMi-2 and various dMi-2-associated proteins and the author's interpretation that Gtsf1 and Piwi biochemically and functionally interact with the dNuRD complex. The revised manuscript contains a greatly expanded interactome and interaction analysis and includes essential negative controls. The new data substantiates the reported interactions but has also changed the authors' conclusions. They no longer propose that Gtsf1/Piwi complex with dNuRD but instead that they complex with an dMi-2/dMEP1 complex and, independently, with dRPD3/p55. This conclusion is now much better supported by the data shown.

My second major concern was the representation and quantification of the experiment shown in Figure 5 (effect of depletion of various proteins on the apiRNA-mediated repression of the krimp gene) which made it difficult for me to interpret the data. In the revised version of the manuscript, the representation of the data is much improved and, as a consequence, the data is much easier to interpret. However, I fail to see a significant effect of dMEP1, dMi-2, dRpd3 and Piwi depletion on apiRNA-mediated repression. The krimp RNA levels after apiRNA expression (Figure 5c, "+ apiRNA" bars) range from approx. 1.0 to 1.4. The fold changes (Figure 5b; krimp RNA levels in presence of apiRNA over control (GFP)) appear more substantial (increasing from approx. 0.4 in the control to a maximum of 1.0 in the siRpd3 experiment) but these changes are largely driven by the apiRNA-independent repression of krimp illustrated in Figure 5c ("- apiRNA" bars). The general problem with this assay is that apiRNA-mediated repression of krimp in the control is already weak (from 1.8 down to 1.1, i.e. less than 2-fold). Given the small effect sizes I find it difficult to disentangle apiRNA-dependent from apiRNA-independent effects (all knockdowns reduce krimp expression even in the absence of apiRNA). The authors write that (line 239) "These data show that these 3 proteins are as important as Piwi for efficient apiRNA-mediated repression in this system." but even Piwi knockdown appears to increase krimp RNA levels in the presence of apiRNA only from 1.1 to 1.3 (Figure 5c)... I do admit that I am not an expert on this type of assay, so my concerns about the significance of the effects might be unfounded. At the very least, the authors should discuss the effect sizes in the text and explain why they represent indeed "significant" (line 238) changes.

We now discuss this apiRNA-independent effect as follows :

"To rule out an apiRNA-independent repression of krimp reporter by these proteins, we checked krimp mRNA levels after MEP-1, Mi-2 and Rpd3 depletion in the absence of apiRNAs. We observed that the knock-down of these proteins caused a slight decrease of krimp mRNA levels in the absence of apiRNAs but not an increase, which would be expected in case of apiRNAs-independent repression (Fig. 5c)."

Minor point:

New dual-luciferase co-immunoprecipitation assays are included. These rely on equal expression of the luciferase fusion proteins. I suggest that the authors document the

expression levels of the various constructs used, in particular in cases where no interaction was detected.

During the reviewing process, we have reproduced the IP LUMIER experiment. We found a similar expression level of the different constructs in the Input. In the final version of Figure 6b we now present the new data set. This set also revealed a decrease of the interaction with the MEP-1 Δ Ct construct. This decrease is not as important as it was in our previous experiment, when the expression of the construct in the Input was too low. We are still obtaining the same conclusion however we modified the text accordingly.

Reviewer #3 (Remarks to the Author):

The authors did a good job improving the manuscript, and now present a much more balanced story. Unfortunately, this also led to a significant down-tuning of the statements on complexes, as the interactions that are newly uncovered do not appear to be found in stable complexes. Of course, this can always be due to experimental conditions. A nice co-elution would have made the story more straightforward, but it is as it is. The question is whether the evidence implicating these new components in piRNA driven silencing is strong enough, and I would say it is. The discussion also nicely places the findings into perspective. The only finding that is not well discussed is the fact that in vivo Rpd3 depletion did not have much of an effect. Hence, for this protein, the evidence that it really acts in the Piwi pathway is significantly weaker. This is not reflected in the discussion.

We agree that, in ovaries, Rpd3 KD has a weaker effect than, for instance, Mi-2 KD. However they both cause similar strong derepressions in OSC cells. The cause of this discrepancy is not discussed but it is now clearly reflected in the revised version of the Discussion as follows :
“Here, we report that a chromatin eraser, Rpd3, and a nucleosome remodeler, Mi-2, in complex with its partner MEP-1, are required for the Piwi-piRNA-dependent TE silencing in OSC cells. The depletion of Mi-2 and MEP-1 also results in TE transcriptional derepression in fly ovaries (Fig. 3 and 4).”